# CAN LLMS ALLEVIATE CATASTROPHIC FORGETTING IN GRAPH CONTINUAL LEARNING? A SYSTEMATIC STUDY

## ABSTRACT

Nowadays, real-world data, including graph-structured data, often arrives in a streaming manner, which means that learning systems need to continuously acquire new knowledge without forgetting previously learned information. Although substantial existing works attempt to address catastrophic forgetting in graph machine learning, they are all based on training from scratch with streaming data. With the rise of pretrained models, an increasing number of studies have leveraged their strong generalization ability for continual learning. Therefore, in this work, we attempt to answer *whether large language models (LLMs) can mitigate catastrophic forgetting in Graph Continual Learning (GCL)*. We first point out that current experimental setups for GCL have significant flaws, as the evaluation stage may lead to task ID leakage. Then, we evaluate the performance of LLMs in more realistic scenarios and find that even minor modifications can lead to outstanding results. Finally, based on extensive experiments, we propose a simple-yet-effective method, Simple Graph Continual Learning (SimGCL), that surpasses the previous state-of-the-art GNN-based baseline by around 20% under the rehearsal-free constraint. To facilitate reproducibility, we have developed an easy-to-use benchmark `LLM4GCL` for training and evaluating existing GCL methods. The code is available at our repository.

## 1 INTRODUCTION

Graph Machine Learning (GML) has become a cornerstone of modern data science, enabling intelligent systems to analyze complex relational structures across domains such as social networks (Fan et al., 2019), financial networks (Han et al., 2025; Li et al., 2022; Yang et al., 2025), recommendation systems (Li et al., 2025b; Wu et al., 2022), and biological structures.(Chen et al., 2023; Wang et al., 2022). Currently, with the remarkable capability to model diverse graph structures and node attributes, Graph Neural Networks (GNNs) have emerged as a prevalent approach for GML tasks. However, most existing approaches assume a static setting where the entire dataset and label space are known in advance. In contrast, real-world graphs are dynamic in nature, as new nodes, edges, and even previously unseen classes may appear over time. This evolving structure introduces the need for Graph Continual Learning (GCL), where models must incrementally integrate new knowledge while retaining previously learned information. Although significant progress has been made in GCL, with numerous studies (Liu et al., 2021; 2023b; Niu et al., 2024a; Zhang et al., 2023a; 2022b;c; 2023b; Zhou & Cao, 2021), substantial challenges still remain unresolved:

- **A Lack of investigation into the rationality of experimental setups in GCL.** This paper focuses specifically on the more mainstream and challenging setting of Node-level Class-Incremental Learning (NCIL). Based on the scope of each task's test set, we divide NCIL into two categories: *local testing* and *global testing*. Although numerous works have been proposed within each category, there is currently no study that analyzes the rationality of the setups.

- **Existing methods are all trained from scratch on streaming graph-structure data.** The great success of pretrained models like GPT (Achiam et al., 2023) and CLIP (Radford et al., 2021) in fields such as natural language processing and computer vision has provided new opportunities

for continual learning. However, due to a lack of powerful foundation models in GML, existing baselines are all trained from scratch.

- **No existing studies that evaluate the performance of LLM-based methods in GCL.** Currently, LLM-based methods have achieved state-of-the-art performance in node classification tasks (Li et al., 2024b; Wu et al., 2025), significantly outperforming traditional GNN models. However, there is a lack of benchmark evaluations assessing the performance of these models in the GCL setting.

To fill these gaps, we systematically analyze the NCIL task and explore the opportunities for mitigating catastrophic forgetting in GCL via LLMs. Specifically, we first identify a flaw in the local testing setting of NCIL, where task ID leakage occurs. This causes class-incremental learning to degrade into task-incremental learning, significantly reducing the task's difficulty. We employ an extremely simple method to predict the task ID with 100% accuracy, achieving a forget-free performance and matching the results of the previous SOTA model (Niu et al., 2024b). Subsequently, we systematically evaluate the performance of LLMs and Graph-enhanced LLMs (GLMs) in global testing, a more realistic and challenging setting. We find that by incorporating common techniques such as Parameter Efficient Fine-Tuning (PEFT) or prototype classifiers, it is possible to leverage the strong generalization ability of pretrained models to surpass existing baseline methods. Finally, based on extensive experiments, we propose a simple yet effective method called SimGCL (Simple Graph Continual Learning for short). Technically, to better incorporate graph structural knowledge, we design ego-graph-derived prompts for each node, which effectively capture textual and structural features of the nodes. Then, in the first session, we applied LoRA (Hu et al., 2022) to efficiently fine-tune the LLMs with instruction tuning, aiming to narrow the distribution gap between the specific dataset and the pretrained dataset. In the incremental sessions, we utilize a training-free prototype classifier for prediction, avoiding parameter updates caused by the arrival of new knowledge, which greatly alleviates catastrophic forgetting. Based on experiments in NCIL and Few-Shot Node-level Class-Incremental Learning (FSNCIL), our proposed SimGCL can significantly surpass the current SOTA models.

In summary, our main contributions are as follows:

- **Analysis of the rationality of experimental setups.** This paper is the first to analyze the flaws in certain experimental setups in GCL and prevent task ID leakage.

- **Comprehensive benchmark.** We propose `LLM4GCL`, the first comprehensive benchmark for LLMs on GCL. We integrate 9 LLM-based methods and 7 textual-attributed graphs for evaluations.

- **Model design.** We propose a simple yet effective model, SimGCL, which is able to surpass all existing baselines and achieve an absolute increase of nearly 20% on certain datasets.

- **Easy-to-use platform.** To facilitate future work and help researchers quickly use the latest models, we develop an easy-to-use open-source platform. We hope to encourage more researchers to explore the use of pretrained models to mitigate catastrophic forgetting in GCL. The code is available at: https://anonymous.4open.science/r/LLM4GCL-ICLR.

## 2 FORMULATIONS AND BACKGROUND

**Text-attributed Graphs (TAGs).** In this benchmark, we evaluate the continual learning capabilities of LLMs through node classification tasks on Text-Attributed Graphs (TAGs) (Ma & Tang, 2021). Formally, a TAG is represented as $\mathcal{G} = (\mathcal{V}, \mathcal{E}, \mathcal{R})$, where $\mathcal{V}$ is the set of $N$ nodes, $\mathcal{E}$ is the edge set, and $\mathcal{R}$ is the collection of textual descriptions associated with each node $v_i \in \mathcal{V}$. Traditionally, the textual attributes of nodes can be encoded into shallow embeddings as $\mathbf{X} = [\mathbf{x}_1, \mathbf{x}_2, ..., \mathbf{x}_N] \in \mathbb{R}^{N \times d}$, where $d$ is the embedding dimension. Such an approach is widely adopted in the graph neural network literature. In contrast, LLM-based approaches directly process the raw text. Given a TAG with labeled nodes $\mathcal{V}^l$, the node classification task aims to predict the labels of the unlabeled nodes $\mathcal{V}^u$.

**Node-Level Class-Incremental Learning (NCIL).** NCIL represents a fundamental paradigm in GCL, addressing incremental node classification tasks in an expanding graph $\mathcal{G}$. As shown in Figure 1, NCIL consists an ordered sequence of training tasks $\{\mathcal{T}_{s_1}, ..., \mathcal{T}_{s_n}\}$ where each task $\mathcal{T}_{s_i} = \{\mathcal{G}_{s_i}, \mathcal{C}_{s_i}, \mathcal{V}^l_{s_i}\}$ consists of a task subgraph $\mathcal{G}_{s_i} \subset \mathcal{G}$, a class set $\mathcal{C}_{s_i}$, and labeled nodes $\mathcal{V}^l_{s_i}$, with $\mathcal{G}_{s_i} \cap \mathcal{G}_{s_j} = \mathcal{C}_{s_i} \cap \mathcal{C}_{s_j} = \varnothing$ and $|\mathcal{C}_{s_i}| = |\mathcal{C}_{s_j}|, |\mathcal{V}^l_{s_i}| = |\mathcal{V}^l_{s_j}|$ for all $i \neq j$. In NCIL, the model is trained sequentially from $\mathcal{T}_{s_1}$ to $\mathcal{T}_{s_n}$. This paradigm well reflects the real-world applications, where

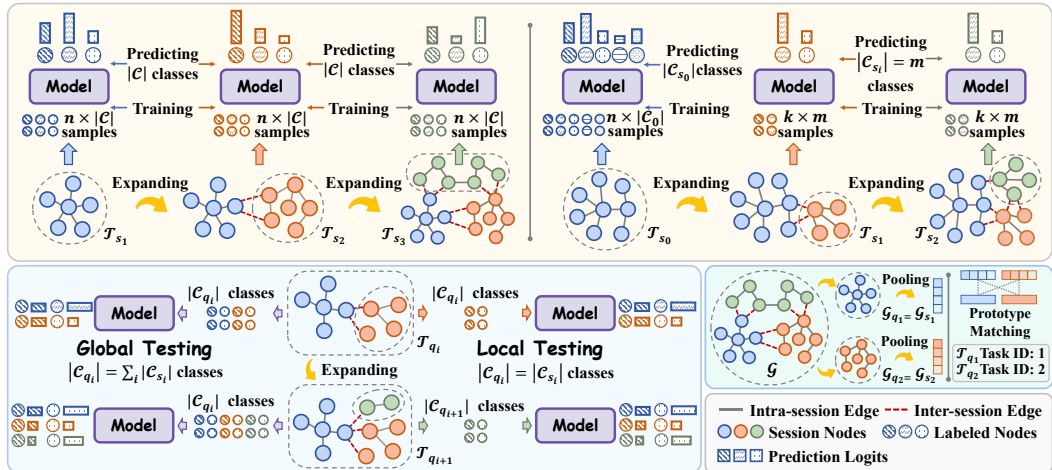

Figure 1: The settings of LLM4GCL. **Upper** : The training pipelines for NCIL (left) and FSNCIL (right), where the model is trained exclusively on the graph of the current task to predict its corresponding classes. **Lower-left** : Two testing methods: *global testing*, which uses the complete graph with inter-session edges, and *local testing*, which evaluates the model on each task-specific graph sequentially. **Lower-right** : The Task ID Leakage issue inherent in local testing, where the model can infer the task ID of a given test graph via pooling and prototype matching.

new categories (e.g., new products for recommendation systems or novel research fields for citation networks) are continually incorporated into the graph in a streaming manner as the domain evolves.

**Few-Shot Node-Level Class-Incremental Learning (FSNCIL).** Newly emerged categories always contain a few samples. Consequently, it is difficult to collect sufficient samples for incremental tasks that match the initial task's scale, leading to the FSNCIL paradigm. Given an ordered sequence of training tasks $\{\mathcal{T}_{s_0}, \mathcal{T}_{s_1}, ..., \mathcal{T}_{s_{n-1}}\}$ where $\mathcal{T}_{s_0}$ serves as the base session and subsequent $\mathcal{T}_{s_i}$ ($i \in \{1, 2, \ldots, n-1\}$) represent incremental sessions, each $\mathcal{T}_{s_i} = \{\mathcal{G}_{s_i}, \mathcal{C}_{s_i}, \mathcal{V}_{s_i}^l\}$ satisfies $\mathcal{G}_{s_i} \cap \mathcal{G}_{s_j} = \mathcal{C}_{s_i} \cap \mathcal{C}_{s_j} = \varnothing$ for all $i \neq j$, and $\mathcal{G}_{s_i} \cap \mathcal{G}_{s_0} = \mathcal{C}_{s_i} \cap \mathcal{C}_{s_0} = \varnothing$. Distinct from NCIL, in FSNCIL, the base session contains both more samples and more classes than incremental sessions which follow a strict $m$-way $k$-shot paradigm, i.e., $|\mathcal{C}_{s_0}| \gg |\mathcal{C}_{s_i}| = |\mathcal{C}_{s_j}| = m$ and $|\mathcal{V}_{s_0}^l| \gg |\mathcal{V}_{s_i}^l| = |\mathcal{V}_{s_j}^l| = k$ for $1 \leq i \leq j$[1]. During training, models are pre-trained on the base session $\mathcal{T}_{s_0}$, followed by fine-tuning sequentially on the incremental session sequence $\mathcal{T}_{s_1} - \mathcal{T}_{s_{n-1}}$.

## 3 LLM4GCL: A SYSTEMATIC STUDY OF LLMS FOR GCL

### 3.1 LEARNING SCENARIOS

LLM4GCL supports both NCIL and FSNCIL paradigms. Prior to this benchmark, existing GCL methods prevalently adopted the evaluation metrics from CGLB (Zhang et al., 2022a), which we refer to as the *local testing*. Specifically, when the $i$-the task $\mathcal{T}_{s_i}$ has been trained and the the model need to be evaluated, this testing manner has an ordered testing task sequence $\{\mathcal{T}_{q_1}, \ldots, \mathcal{T}_{q_i}\}$, where $i \leq n$. The evaluation process consists of $i$ independent evaluations, with each testing task $\mathcal{T}_{q_j} = \{\mathcal{G}_{q_j}, \mathcal{C}_{q_j}\}$ ($j \leq i$) leveraging the local subgraph ($\mathcal{G}_{q_j} = \mathcal{G}_{s_j}$) and global class set ($\mathcal{C}_{q_j} = \bigcup_{z=1}^{j} \mathcal{C}_{s_z}$), respectively. However, we highlight that this setup has a fundamental flaw: **for each task, since testing and training samples are drawn from the same subgraph, their distribution similarity makes it trivial to accurately predict task ID, degrading class-incremental to task-incremental learning.** For instance, TPP (Niu et al., 2024b) transductively captures task-specific prototypes utilizing a Laplacian smoothing-based matching approach, achieving 100% task ID prediction accuracy and 0% forgetting ratio. However, as demonstrated in Table 1, our analysis reveals that even using a basic

---

[1]FSNCIL typically features more classes in its base session than NCIL. Conversely, NCIL includes more classes and samples than the incremental sessions of FSNCIL. For a direct comparison of these settings, see Appendix B.1; detailed experimental configurations are provided in Appendix B.3, Table 6.

Table 1: Performance evaluation on three pipelines: (1) GNN for feature aggregation + TPP (Laplacian smoothing) for prototype generation; (2) GNN aggregation + standard mean pooling (MP) for prototypes; (3) MLP aggregation + MP for prototypes. AA and AF represent average accuracy and forgetting ratio across all sessions.

| Backbone | Method | Cora | | Citeseer | | WikiCS | | Photo | | Products | | Arxiv-23 | | Arxiv | |
|---|---|---|---|---|---|---|---|---|---|---|---|---|---|---|---|
| | | AA | AF | AA | AF | AA | AF | AA | AF | AA | AF | AA | AF | AA | AF |
| GNN | TPP | 95.2 | -0.0 | 87.9 | -0.0 | 93.6 | -0.0 | 91.3 | -0.0 | 89.6 | -0.0 | 91.6 | -0.0 | 85.7 | -0.0 |
| GNN | MP | 95.2 | -0.0 | 87.9 | -0.0 | 93.6 | -0.0 | 91.3 | -0.0 | 89.6 | -0.0 | 91.6 | -0.0 | 85.7 | -0.0 |
| MLP | MP | 90.3 | -0.0 | 86.9 | -0.0 | 89.2 | -0.0 | 79.0 | -0.0 | 83.1 | -0.0 | 88.8 | -0.0 | 81.6 | -0.0 |

mean pooling operation to get prototypes achieves flawless task ID prediction. Furthermore, with accurate task identification, the model can also attain substantial performance by discarding the GNN entirely and simply employing task-specific two-layer MLPs for classification[2]. Consequently, we argue that this evaluation method is fundamentally flawed, as it fails to accurately assess the model's genuine continual learning capability. In this paper, we primarily investigate the more challenging and realistic *global testing* setup. In *global testing*, we also have an ordered testing task sequence, but the difference is that for each testing task $\mathcal{T}_{q_j} = \{\mathcal{G}_{q_j}, \mathcal{C}_{q_j}\}$ ($j \leq i$), the evaluated graph is the union of the subgraphs from all previous tasks ($\mathcal{G}_{q_j} = \bigcup_{z=1}^{j} \mathcal{G}_{s_z}$) and the class set is still global ($\mathcal{C}_{q_j} = \bigcup_{z=1}^{j} \mathcal{C}_{s_z}$). This setup not only better reflects real-world scenarios but also prevents the issue of task ID leakage, enabling authentic assessment of continual learning performance[3].

In addition to evaluation settings, *previous knowledge leakage* and *label imbalance* also constitute significant concerns for both NCIL and FSNCIL approaches. Since GNNs' message-passing mechanisms inherently aggregate neighborhood information, these edges connecting old and new task nodes potentially expose previous task knowledge during new task training (Zhang et al., 2022a). Given that real-world scenarios often prohibit access to previous task data due to privacy or storage constraints, our benchmark excludes inter-task edges, using only intra-task connections. As for the label imbalance issue, since each session contains only a limited number of classes, while sample sizes vary substantially across classes, the imbalance issue becomes pronounced in continual learning settings. To prevent performance degradation caused by this issue, our benchmark removes the classes with insufficient samples and utilizes a unified sample size for each remaining class.

## 3.2 BENCHMARK SETTINGS

**Datasets.** To ensure comprehensive evaluation, we gather seven diverse and representative datasets: Cora (He et al., 2023), Citeseer (Chen et al., 2024b), WikiCS (Liu et al., 2023a), Photo (Yan et al., 2023), Products (Hu et al., 2020), Arxiv-23 (He et al., 2023), and Arxiv (Hu et al., 2020), according to the following criteria: (1) **Multiple domains.** LLM4GCL incorporates datasets spanning multiple domains, including citation networks, web link networks, and e-commerce networks, reflecting diverse real-world applications; (2) **Diverse scale and density.** LLM4GCL datasets cover a wide range of scales, from thousands to hundreds of thousands of nodes. Moreover, the density of these datasets also varies significantly, supporting a robust evaluation of the model's effectiveness; (3) **Various sessions.** LLM4GCL datasets provide varied session settings of both total session number and classes per session, allowing comprehensive analysis of the model's continual learning performance. The statistics of these datasets and descriptions are provided in Appendix B.3.

**Baselines.** LLM4GCL evaluates three baseline categories differentiated by backbone integration strategies: (1) **GNN-based methods**, including Vanilla GCN (Kipf & Welling, 2016), EWC (Kirkpatrick et al., 2017), LwF (Li & Hoiem, 2017), Cosine, and TPP (Niu et al., 2024b) for NCIL, with TEEN (Wang et al., 2023a) added for FSNCIL; (2) **LLM-based methods**, including encoder-only BERT (Devlin et al., 2019) and RoBERTa (Liu et al., 2019), decoder-only LLaMA (Touvron et al., 2023), and RoBERTa integrated with SimpleCIL (Zhou et al., 2025); (3) **GLM-based methods**, including two LLM-as-Enhancer method $GCN_{Emb}$ (Wu et al., 2025) and ENGINE (Zhu et al., 2024), and three LLM-as-Predictor approaches GraphPrompter (Liu et al., 2024), GraphGPT (Tang et al., 2024a), and LLaGA (Chen et al., 2024a). Extended descriptions are provided in Appendix B.4 and C.

---

[2]Smoothing and mean pooling are applied to all nodes of each test graph $\mathcal{G}_{q_i}$, as shown in Figure 1.

[3]For comprehensive discussion on the two test settings, and the details of TPP, please see Appendix B.2.

**Metrics.** In `LLM4GCL`, we employ Accuracy as the evaluation metric. Following established continual learning benchmark (Rebuffi et al., 2017), we define $\mathcal{A}_i$ as the accuracy after the $i$-th training task. The final metrics for both NCIL and FSNCIL are $\bar{\mathcal{A}} = \frac{1}{n}\sum_{i=1}^{n} \mathcal{A}_i$ and $\mathcal{A}_N$, which refer to the average accuracy across all tasks and the accuracy after the last task, respectively. These two metrics can well assess the model's overall performance and ultimate knowledge preservation.

### 3.3 SIMGCL: A SIMPLE YET EFFECTIVE GLM-BASED APPROACH FOR GCL

Currently, the application of LLMs and GLMs to GCL tasks remains nascent. To the best of our knowledge, there are no existing LLM- or GLM-based methods specifically designed for GCL tasks. While directly employing GLMs offers a straightforward solution by combining evolving graph structures with LLMs' generalization capabilities, this approach incurs substantial computational overhead. Furthermore, as new tasks accumulate, LLMs' general knowledge progressively degrades, resulting in a biased preference toward newer tasks. Consequently, we propose SimGCL, as depicted in Figure 2, which demonstrates two advantages: (1) **Efficiency**. SimGCL only requires a single round of instruction tuning in the first session, dramatically reducing training costs. (2) **Generalization**. SimGCL

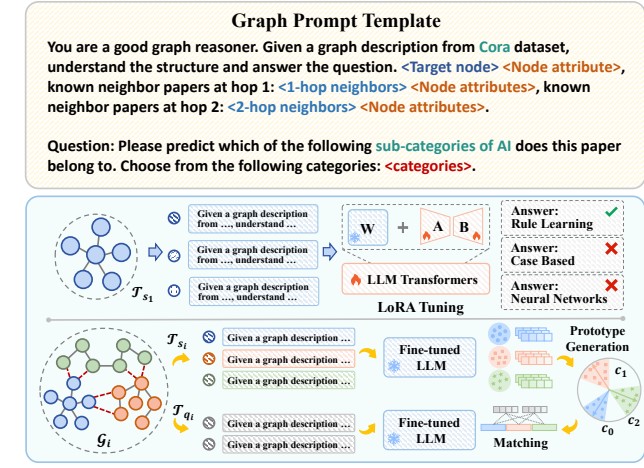

Figure 2: Proposed SimGCL framework, which executes two stages: (1) fine-tuning with graph prompts and instructions, followed by (2) training-free prototype generation and classification using LLM-generated node embeddings.

constructs class prototypes directly from the embeddings produced by its fine-tuned LLM. This training-free approach avoids parameter updates, thereby mitigating catastrophic forgetting.

**Instruction Tuning.** The instruction tuning phase facilitates LLM adaptation to graph-structured tasks by bridging the distributional gap between the pretraining corpus and target task datasets. Inspired by (Wang et al., 2025), we employ a neighborhood-aware graph prompt to guide the LLM's processing. Derived from each node's ego-graph $\tilde{\mathcal{G}}_v$, this prompt template comprises three components: (1) Task description outlining the scenario and dataset; (2) Graph structure incorporating target node attributes and neighborhood structure, with each node represented in the format of `[Node ID][Node Text]`; and (3) Question that specifies requirements and label text options. Moreover, to ensure training efficiency, we limit LLM tuning to the first session and adopt LoRA (Hu et al., 2022) instead of full-parameter fine-tuning to prevent overfitting.

**Prototype Generation and Classification.** Following instruction tuning, SimGCL progressively generates class prototypes across sessions. Formally, let $f_\theta : (r_{\text{prompt}}, \mathcal{G}_v) \mapsto \mathbf{h}_v$ denote the LLM's mapping from inputs to final-layer embeddings, the prototype $\mathbf{c}_i$ for class $i$ is computed as:

$$\mathbf{c}_i = \frac{1}{K}\sum_{j=1}^{|\mathcal{Y}_b|} \mathbb{I}(y_j = i)\mathbf{h}_j \tag{1}$$

where $K = \sum_{j=1}^{|\mathcal{Y}_b|} \mathbb{I}(y_j = i)$, with $|\mathcal{Y}_b|$ being the number of labeled nodes in session $b$ where class $i$ appears, and $\mathbb{I}$ denotes the indicator function. Then during prediction, for node $v_i$ with its embedding $\mathbf{h}_i$, its probability of belonging to class $j$ is formulated as follows:

$$\hat{y}_{i,j} = \frac{\mathbf{h}_i \cdot \mathbf{c}_j}{\|\mathbf{h}_i\| \cdot \|\mathbf{c}_j\|} \cdot \tau \tag{2}$$

where $\tau(\tau > 0)$ is the scaling hyperparameter controlling the weight distribution. This training-free approach effectively preserves class-discriminative features across sessions, simultaneously reducing computational cost for new sessions while alleviating catastrophic forgetting in LLMs.

Table 2: Performance comparison of GNN-, LLM-, and GLM-based methods in the NCIL scenario of `LLM4GCL`. The best and second results are highlighted.

| Methods | Cora | | Citeseer | | WikiCS | | Photo | | Products | | Arxiv-23 | | Arxiv | |
|---|---|---|---|---|---|---|---|---|---|---|---|---|---|---|
| | $\bar{A}$ | $\mathcal{A}_N$ | $\bar{A}$ | $\mathcal{A}_N$ | $\bar{A}$ | $\mathcal{A}_N$ | $\bar{A}$ | $\mathcal{A}_N$ | $\bar{A}$ | $\mathcal{A}_N$ | $\bar{A}$ | $\mathcal{A}_N$ | $\bar{A}$ | $\mathcal{A}_N$ |
| GCN | 57.0 | 38.2 | 52.4 | 30.2 | 54.9 | 34.9 | 46.5 | 19.9 | 25.5 | 5.4 | 19.9 | 2.9 | 21.2 | 1.5 |
| EWC | 56.0 | 31.0 | 45.9 | 28.5 | 55.3 | 33.0 | 46.9 | 20.5 | 29.4 | 15.9 | 25.7 | 15.0 | 24.9 | 13.8 |
| LwF | 55.7 | 30.8 | 47.8 | 28.2 | 55.0 | 34.0 | 45.8 | 20.4 | 26.0 | 7.5 | 21.1 | 6.8 | 18.9 | 1.6 |
| Cosine | 65.4 | 45.2 | 50.7 | 31.0 | 66.5 | 53.5 | 63.6 | 49.6 | 36.1 | 16.1 | 36.1 | 24.2 | 37.4 | 27.8 |
| TPP | 45.7 | 13.7 | 45.4 | 9.6 | 35.1 | 9.8 | 36.3 | 5.7 | 15.0 | 0.0 | 12.2 | 0.3 | 19.6 | 3.4 |
| BERT | 56.0 | 29.9 | 53.8 | 28.7 | 58.4 | 30.0 | 43.4 | 18.4 | 26.9 | 4.1 | 27.1 | 5.1 | 26.0 | 9.7 |
| RoBERTa | 54.6 | 29.6 | 54.1 | 28.6 | 55.1 | 30.8 | 43.5 | 19.1 | 27.6 | 3.2 | 23.1 | 0.8 | 24.4 | 1.4 |
| LLaMA | 65.6 | 53.8 | 55.7 | 31.7 | 55.5 | 30.9 | 44.6 | 19.1 | 29.8 | 0.4 | 24.3 | 1.0 | 24.9 | 1.4 |
| SimpleCIL | 70.8 | 58.3 | 66.4 | 49.5 | 71.4 | 57.3 | 62.1 | 52.5 | 66.8 | 52.6 | 52.4 | 38.8 | 50.6 | 36.5 |
| GCN$_{\text{LLMEmb}}$ | 59.1 | 31.1 | 53.6 | 30.4 | 53.4 | 27.5 | 47.7 | 21.0 | 26.9 | 0.1 | 22.9 | 0.8 | 24.3 | 1.4 |
| ENGINE | 59.2 | 31.3 | 53.5 | 29.8 | 56.4 | 30.1 | 47.9 | 21.0 | 27.2 | 1.1 | 22.5 | 0.7 | 24.6 | 1.3 |
| GraphPrompter | 61.9 | 46.8 | 60.2 | 30.6 | 59.6 | 38.3 | 51.5 | 31.0 | 29.0 | 0.8 | 23.4 | 0.9 | 24.8 | 1.4 |
| GraphGPT | 55.5 | 31.6 | 60.0 | 30.1 | 62.0 | 49.2 | 50.8 | 30.2 | 35.5 | 3.2 | 30.7 | 5.8 | 32.2 | 3.9 |
| LLaGA | 58.2 | 30.2 | 51.3 | 27.8 | 53.7 | 27.6 | 47.2 | 20.7 | 25.7 | 0.2 | 26.9 | 4.1 | 26.1 | 1.3 |
| SimGCL (Ours) | 84.6 | 80.0 | 77.1 | 66.3 | 73.5 | 61.9 | 82.1 | 72.6 | 71.1 | 60.2 | 38.7 | 13.6 | 59.9 | 33.8 |

## 4 EXPERIMENTS AND ANALYSIS

In this section, we conduct extensive experiments to analyze the performance of various methods on GCL tasks. Specifically, Tables 2 and 3 show the results for the NCIL and FSNCIL scenarios, respectively. Figure 3 compares the influence of various LLM model sizes, while Figure 4 shows how the performance changes as the number of sessions increases. Additional experiments, including results on domain-incremental learning, time-efficiency analysis, and visualizations, are available in Appendix E. Based on these results, we give the following key observations (**Obs.**):

**Obs. ❶ GNN-based methods exhibit persistent limitations in both NCIL and FSNCIL scenarios.** As shown in Table 2 and Table 3, GNN-based approaches consistently underperform across all seven datasets. Notably, even on the small-scale Cora and Citeseer datasets, their final incremental accuracy ($\mathcal{A}_N$) fails to exceed 50%. This challenge even escalates significantly in larger-scale datasets like Arxiv-23 and Arxiv, where GCN and LwF achieve merely 1.5% and 1.6%, 2.0% and 3.8% accuracy after the final session of NCIL and FSNCIL, respectively, underscoring severe catastrophic forgetting. Moreover, classic CL methods (e.g., EWC and LwF) demonstrate inferior performance to vanilla GCN on shorter-session datasets and yield only marginal improvements in longer-session scenarios, while TPP fails to accurately identify task IDs, resulting in severely degraded performance under *global testing* settings. These limitations underscore the critical need for novel approaches to address GCIL and GFSCIL challenges effectively.

**Obs. ❷ LLMs demonstrate higher performance in both NCIL and FSNCIL tasks, even without explicit graph structure utilization.** As shown in Table 2 and Table 3, while the encoder-only BERT and RoBERTa achieve performance only comparable to or marginally better than GCN, the decoder-only LLaMA attains a 16.9% improvement on Citeseer in the FSNCIL scenario, while SimpleCIL surpasses all the GNN-based models in most datasets and metrics (24 out of 28), with average improvements of 12.93% in NCIL and 9.7% in FSNCIL, respectively. We attribute this superior performance to the LLM backbone's exceptional generalization capability for processing textual features and capturing class-specific semantics. The results also demonstrate the substantial potential of LLMs in addressing NCIL and FSNCIL challenges.

**Obs. ❸ Current GLM-based methods demonstrate unsatisfactory performance on both NCIL and FSNCIL scenarios.** Contrary to expectations, deliberately designed GLMs fail to achieve competitive performance, indicating that current GLM approaches exhibit greater limitations in GCL compared to pure LLM-based methods. We subsequently analyze this phenomenon through two aspects: ① **LLM-as-Enhancer.** GCN$_{\text{Emb}}$ and ENGINE employ LLMs to augment graph features, yielding minimal but consistent NCIL improvements (0.2%–3.4% over vanilla GCN). However, in

Table 3: Performance comparison of GNN-, LLM-, and GLM-based methods in the FSNCIL scenario of `LLM4GCL`. The best and second results are highlighted.

| Methods | Cora | | Citeseer | | WikiCS | | Photo | | Products | | Arxiv-23 | | Arxiv | |
|---|---|---|---|---|---|---|---|---|---|---|---|---|---|---|
| | $\bar{\mathcal{A}}$ | $\mathcal{A}_N$ | $\bar{\mathcal{A}}$ | $\mathcal{A}_N$ | $\bar{\mathcal{A}}$ | $\mathcal{A}_N$ | $\bar{\mathcal{A}}$ | $\mathcal{A}_N$ | $\bar{\mathcal{A}}$ | $\mathcal{A}_N$ | $\bar{\mathcal{A}}$ | $\mathcal{A}_N$ | $\bar{\mathcal{A}}$ | $\mathcal{A}_N$ |
| GCN | 68.0 | 38.1 | 39.5 | 17.4 | 62.4 | 47.4 | 58.5 | 32.3 | 36.0 | 14.1 | 22.2 | 5.4 | 25.3 | 2.0 |
| EWC | 59.0 | 36.3 | 49.2 | 21.2 | 58.4 | 40.4 | 62.0 | 28.4 | 45.7 | 31.5 | 36.4 | 29.5 | 31.5 | 18.3 |
| LwF | 63.3 | 43.5 | 45.1 | 20.7 | 59.4 | 41.0 | 60.1 | 29.4 | 50.3 | 38.7 | 30.8 | 15.1 | 27.8 | 3.8 |
| Cosine | 72.6 | 57.8 | 49.1 | 25.7 | 68.0 | 50.7 | 67.9 | 50.5 | 50.9 | 33.6 | 40.1 | 27.9 | 27.2 | 17.2 |
| TEEN | 60.9 | 40.3 | 59.0 | 39.5 | 59.3 | 42.4 | 59.3 | 35.5 | 49.6 | 28.4 | 39.2 | 27.1 | 31.8 | 18.6 |
| TPP | 39.0 | 9.1 | 37.3 | 12.6 | 37.3 | 12.4 | 40.0 | 14.0 | 14.0 | 0.0 | 7.3 | 0.0 | 11.1 | 6.0 |
| BERT | 56.4 | 34.7 | 61.1 | 38.8 | 61.7 | 33.0 | 47.7 | 25.8 | 22.4 | 3.8 | 15.2 | 3.4 | 14.9 | 6.0 |
| RoBERTa | 59.6 | 41.9 | 54.0 | 29.2 | 67.2 | 42.3 | 58.2 | 29.6 | 38.8 | 6.9 | 22.6 | 1.4 | 25.7 | 1.3 |
| LLaMA | 72.6 | 55.6 | 75.9 | 55.5 | 65.2 | 43.9 | 61.4 | 32.3 | 43.5 | 13.7 | 23.5 | 1.5 | 22.7 | 1.4 |
| SimpleCIL | 69.6 | 53.6 | 64.1 | 49.9 | 73.2 | 63.1 | 66.3 | 53.0 | 65.6 | 53.6 | 49.8 | 40.0 | 46.4 | 36.6 |
| GCN$_{\text{LLMEmb}}$ | 68.2 | 40.1 | 54.3 | 28.7 | 54.7 | 31.0 | 66.0 | 34.2 | 30.6 | 0.2 | 21.4 | 1.0 | 22.1 | 0.9 |
| ENGINE | 52.2 | 28.3 | 47.6 | 25.5 | 46.8 | 23.7 | 48.0 | 21.5 | 20.9 | 0.1 | 17.4 | 0.5 | 17.3 | 1.4 |
| GraphPrompter | 63.2 | 37.3 | 65.3 | 34.5 | 69.5 | 51.6 | 69.7 | 51.0 | 37.9 | 2.2 | 27.4 | 2.3 | 27.3 | 1.0 |
| GraphGPT | 62.4 | 39.6 | 65.0 | 41.7 | 71.2 | 61.6 | 62.2 | 38.9 | 43.2 | 16.2 | 25.4 | 1.7 | 24.3 | 2.0 |
| LLaGA | 62.0 | 39.6 | 52.2 | 28.0 | 49.4 | 30.4 | 48.8 | 18.0 | 28.0 | 0.2 | 18.2 | 0.9 | 16.6 | 1.6 |
| SimGCL (Ours) | 78.0 | 67.6 | 78.0 | 63.8 | 68.8 | 64.1 | 81.2 | 71.3 | 69.7 | 62.7 | 31.8 | 10.3 | 36.3 | 6.8 |

FSNCIL settings, these methods exhibit significant performance degradation across WikiCS, Products, Arxiv-23, and Arxiv datasets, underperforming vanilla GCN. We attribute this phenomenon to these models' tendency to overfit training samples, which is further exacerbated by FSNCIL's few-shot learning scenario, ultimately amplifying catastrophic forgetting. These methods still fundamentally rely on GNN architectures for final predictions, inheriting their limitations in continual learning scenarios. Given that the GNN's limited generalization capacity serves as the performance bottleneck, we argue that LLM-as-Enhancer GLMs are suboptimal for GCL scenarios. ② **LLM-as-Predictor.** Among GraphPrompter, GraphGPT, and LLaGA, GraphPrompter demonstrates superior performance, consistently exceeding GCN$_{\text{Emb}}$ across most datasets and metrics. Nevertheless, its advantages over LLaMA remain limited to specific datasets. We attribute this to two key factors: On the one hand, the architectural gap between shallow GNNs and deep LLMs leads to divergent parameter updates during continual learning, worsening inter-modal misalignment, and degrading GLM performance. On the other hand, GLMs like LLaGA possess strong fitting capabilities, which may lead to over-adaptation to recent tasks, consequently compromising their generalization and transfer abilities.

**Obs. ❹ Dense graph structures may enhance GLM effectiveness in GCL.** Table 2 and Table 3 reveal that LLM-as-Predictor GLMs achieve notable performance gains on the WikiCS and Photo datasets, with GraphPrompter and GraphGPT ranking as runner-up or third-best models. This improvement likely stems from these datasets' higher graph density than datasets like Cora or Products, providing richer structural information for LLM to better understand the task scenarios and make predictions. However, all GLMs exhibit suboptimal performance on Arxiv despite their comparable edge density (6.89 edges/node), suggesting that extended session ranges may disproportionately compromise LLMs' generalization capacity, outweighing potential structural benefits.

**Obs. ❺ Prototype-based learning improves cross-task generalization for GNNs and LLMs.** Table 2 and Table 3 demonstrate that prototype-based methods achieve superior performance, with Cosine and SimpleCIL emerging as optimal approaches for GNN- and LLM-based methods, respectively. In NCIL settings with abundant per-session samples, these methods outperform other baselines by significant margins (maximum improvements of 29.7% and 39.2% versus GCN and RoBERTa). This advantage stems from an optimal balance between generalization and plasticity: Initial session tuning bridges the distribution gap between the initialized or pretrained representations and target tasks, while subsequent session prototype generation preserves model parameters, maintaining generalization without catastrophic interference. Moreover, the prototype framework requires only first-session training, offering significant computational efficiency benefits.

**Obs. ❻ SimGCL consistently overperform GNN-, LLM- and GLM-based baselines.** From Table 2 and Table 3, it is obvious that SimGCL consistently overperform other baselines (23 out

Table 4: Performance comparison of `LLM4GCL` baselines in NCIL scenario across different class sizes (W) and session numbers (S) on Arxiv. The best and second results are highlighted.

| Methods | 8W5S | | 5W8S | | 4W10S | | 2W20S | |
|---|---|---|---|---|---|---|---|---|
| | $\bar{\mathcal{A}}$ | $\mathcal{A}_N$ | $\bar{\mathcal{A}}$ | $\mathcal{A}_N$ | $\bar{\mathcal{A}}$ | $\mathcal{A}_N$ | $\bar{\mathcal{A}}$ | $\mathcal{A}_N$ |
| GCN | 24.5 ↓0.0 | 2.0 ↓0.0 | 20.5 ↓0.0 | 1.1 ↓0.0 | 21.2 ↓0.0 | 1.5 ↓0.0 | 13.4 ↓0.0 | 0.7 ↓0.0 |
| EWC | 31.6 ↑7.1 | 19.6 ↑17.6 | 26.2 ↑5.7 | 13.8 ↑12.7 | 24.9 ↑3.7 | 13.8 ↑12.3 | 15.4 ↑2.0 | 4.2 ↑3.5 |
| LwF | 29.2 ↑4.7 | 8.6 ↑6.6 | 22.6 ↑2.1 | 2.7 ↑1.6 | 18.9 ↓2.3 | 1.6 ↑0.1 | 14.2 ↑0.8 | 1.0 ↑0.3 |
| Cosine | 35.6 ↑11.1 | 23.2 ↑21.2 | 38.3 ↑17.8 | 27.9 ↑26.8 | 37.4 ↑16.2 | 27.8 ↑26.3 | 41.6 ↑28.2 | 31.4 ↑30.7 |
| RoBERTa | 33.9 ↑9.4 | 3.9 ↑1.9 | 26.7 ↑6.2 | 2.1 ↑1.0 | 24.4 ↑3.2 | 1.4 ↓0.1 | 15.6 ↑2.2 | 0.3 ↓0.4 |
| LLaMA | 35.0 ↑10.5 | 3.8 ↑1.8 | 27.6 ↑7.1 | 1.5 ↑0.4 | 24.9 ↑3.7 | 1.4 ↓0.1 | 15.9 ↑2.5 | 0.3 ↓0.4 |
| SimpleCIL | 46.1 ↑21.6 | 31.4 ↑29.4 | 49.7 ↑29.2 | 35.9 ↑34.8 | 50.6 ↑29.4 | 36.5 ↑35.0 | 52.6 ↑39.2 | 39.1 ↑38.4 |
| GCN$_{LLMEmb}$ | 33.7 ↑9.2 | 3.8 ↑1.8 | 26.6 ↑6.1 | 2.0 ↑0.9 | 24.3 ↑3.1 | 1.4 ↓0.1 | 15.6 ↑2.2 | 0.3 ↓0.4 |
| ENGINE | 34.1 ↑9.6 | 3.7 ↑1.7 | 26.5 ↑6.0 | 2.0 ↑0.9 | 24.6 ↑3.4 | 1.3 ↓0.2 | 15.7 ↑2.3 | 0.2 ↓0.5 |
| GraphPrompter | 35.1 ↑10.6 | 2.9 ↑0.9 | 27.8 ↑7.3 | 2.0 ↑0.9 | 24.8 ↑3.6 | 1.4 ↓0.1 | 16.8 ↑3.4 | 0.4 ↓0.3 |
| SimGCL (Ours) | 51.6 ↑27.1 | 28.7 ↑26.7 | 53.0 ↑32.5 | 30.4 ↑29.3 | 59.9 ↑38.7 | 33.8 ↑32.3 | 57.4 ↑44.0 | 17.5 ↑16.8 |

of 28), with a maximum 21.7% and 18.0% improvement in NCIL and FSNCIL, respectively. The effectiveness of SimGCL stems from two key factors: (1) its graph-structured instruction tuning and prompting framework enhances LLMs' comprehension of graph topology, yielding higher-quality prototypes enriched with task-specific structural information for more discriminative representations; (2) unlike GraphPrompter and GraphGPT that rely on GNN-derived representations, SimGCL encodes graph information through textual prompts that better align with LLMs' native input space, eliminating cross-architecture representation mismatches. However, SimGCL demonstrates relatively inferior performance on the arxiv-23 dataset and in FSNCIL compared to NCIL, which we attribute to two factors: (1) The sparse graph structure of Arxiv-23 provides limited topological information, potentially compromising LLMs' structural comprehension (Zhu et al., 2024); (2) SimGCL's expanded tuning set (12 classes vs. 4 classes in NCIL) in FSNCIL may promote overfitting to the initial session data, thereby diminishing generalization capability for subsequent incremental sessions.

**Obs. ❼ Scaling LLM parameters enhances generalization.** Figure 3 compares the performance of SimpleCIL and SimGCL across LLM backbones of varying parameter scales. The results indicate that increased parameter counts consistently enhance generalization capacity across both encoder-only and decoder-only architectures, consequently boosting GCL performance throughout all task sessions.

**Obs. ❽ LLMs and GLMs are better at dealing with tasks with short-range sessions.** Table 4 evaluates model performance on the Arxiv dataset across varying session and class configurations[2]. In the 8-way 5-session setting, most LLM- and GLM-based methods achieve performance comparable to or exceeding top GNN-based approaches. However, this advantage diminishes with increasing session numbers, where performance differences between LLM/GLM and GNN methods become negligible. We attribute this convergence to progressive overfitting during incremental sessions, which increasingly weakens the pretrained models' generalization capacity, ultimately reducing their performance advantage. Furthermore, prototype-based methods (i.e., Cosine, SimpleCIL, and SimGCL) demonstrate consistent performance stability across all experimental configurations, as their training-free prototype generation mechanism remains unaffected by session variations. This property makes them particularly suitable for long-session GCL applications.

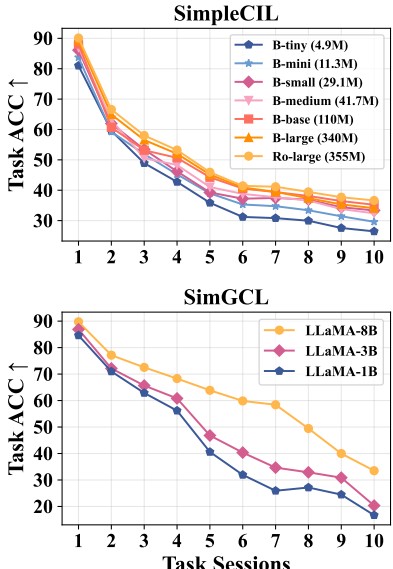

Figure 3: The performance of SimpleCIL and SimGCL across varying LLM backbone sizes on Arxiv. The 'B' and 'Ro' refer to BERT and RoBERTa.

---

[2]Detailed results for BERT, GraphGPT, and LLaGA are provided in Appendix E.

## 5 RELATED WORKS

**Continual Learning with Pre-trained Models (PTMs).** The application of PTMs to continual learning was initially explored in Computer Vision (CV) (Zhou et al., 2025; 2024) and Natural Language Processing (NLP) (Wang et al., 2023b; Zhao et al., 2024b), with Vision Transformer (ViT) (Dosovitskiy et al., 2020) and BERT (Devlin et al., 2019) serving as representative implementations in each domain respectively. Pre-trained on extensive corpora (Brown et al., 2020) rather than trained from scratch, PTMs exhibit inherent advantages in generalizability (Cao et al., 2023), which supports flexible algorithm design, including both frozen-backbone approaches in a non-continual manner (e.g., training-free prototypes (Zhou et al., 2025)) and parameter-efficient tuning (PEFT) methods that preserve pre-trained weights (e.g., prompt tuning (Jia et al., 2022) and adapter learning (Chen et al., 2022)). Furthermore, leveraging the lightweight nature of PEFT blocks, there are also methods that employ orthogonal optimization via auxiliary losses (Wang et al., 2023b) or null-space projection (Lu et al., 2024) to minimize cross-task interference.

**Graph Continual Learning (GCL).** While continual learning in CV and NLP focuses on preventing catastrophic forgetting of semantic representations, GCL must additionally address the forgetting of learned topological patterns induced by evolving graph structures (Zhang et al., 2024). Typically, GCL methods can be categorized based on their approach to mitigating catastrophic forgetting, including (1) regularizing parameter updates through explicit constraints, such as topology weight (Liu et al., 2021) or knowledge distillation (Hoang et al., 2023; Li & Hoiem, 2017); (2) isolating task-critical parameters, typically implemented through expanding parameters or model components to accommodate new knowledge (Zhang et al., 2023a; 2022b); and (3) replaying representative historical data, including key nodes (Zhou & Cao, 2021), sparsified computational subgraphs (Zhang et al., 2022c; 2023b), and condensed task-specific graphs (Liu et al., 2023b; Niu et al., 2024a). However, practical constraints like privacy concerns and storage limitations often prevent access to historical task data, thereby restricting the applicability of replay-based methods. Therefore, our benchmark primarily focuses on replay-free approaches.

**Graph-enhanced Language Models (GLMs).** Recently, with the rapid development of LLMs (e.g., RoBERTa (Conneau et al., 2019), LLaMA (Touvron et al., 2023), GPTs (Achiam et al., 2023)), research interest in leveraging LLMs for graph tasks has grown substantially, resulting in numerous GLM approaches. Generally, these methods can be categorized into three main approaches based on how LLMs interact with graph structures or GNNs (Jin et al., 2024; Li et al., 2023): (1) LLM-as-Enhancer (He et al., 2023; Liu et al., 2023a; Zhu et al., 2024), which leverages LLMs to retrieve node-relevant semantic knowledge, enhancing GNNs' initial embedding quality; (2) LLM-as-Predictor (Chen et al., 2024a; Huang et al., 2024; Li et al., 2024a), which employs LLMs to autoregressively make predictions using graph-structured information, including structural prompts and GNN-encoded representations; and (3) GNN-LLM Alignment (Jin et al., 2023; Zhao et al., 2022), which aligns LLMs with GNNs in the same vector space, enabling GNNs to be more semantically aware. Considering the generalization and semantic understanding capabilities of LLMs, combined with the graph structure modeling capacity of GNNs and other integrations, GLMs are anticipated to provide a novel approach to more effectively addressing GCL tasks.

## 6 CONCLUSION

This paper introduces `LLM4GCL`, the first comprehensive benchmark for evaluating GCL performance of LLMs and graph-enhanced LLMs. Our work makes three key contributions: Firstly, we systematically analyze the current GCL evaluation paradigm, identifying its critical flaws that may cause task ID leakage and compromise performance assessment. (2) We evaluate 9 representative LLM-based methods across 7 textual-attributed graphs, revealing that existing GLM-based approaches underperform in both NCIL and FSNCIL scenarios due to overfitting tendencies and LLM-GNN representation misalignment. However, prototype-based modifications enable these models to achieve state-of-the-art performance, with additional gains attainable through model scaling. (3) We propose SimGCL, an effective GLM-based approach combining graph-prompted instruction tuning with prototype networks, which enhances topological understanding while preserving LLMs' generalization capacity. We believe the proposal of `LLM4GCL` will establish a foundation for future research by delineating effective pathways for PTM-graph integration. Moreover, we have made our code publicly available and welcome further contributions of new datasets and methods.

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

## A   THE USE OF LARGE LANGUAGE MODELS

This work employs large language models for writing assistance and manuscript refinement.

## B   ADDITIONAL DETAILS ON LLM4GCL

### B.1   COMPARISON BETWEEN NCIL AND FSNCIL

In this section, we will give a direct comparison between the two in Table 5.

Table 5: Comparison of NCIL and FSNCIL settings.

| Setting | Session Type | Samples per Class | Classes per Session |
|---------|-------------|-------------------|---------------------|
| NCIL | - | Sufficient (e.g., 200 samples) | Small (e.g., 3 classes) |
| FSNCIL | Base
Incremental | Sufficient (e.g., 200 samples)
Few-shot (e.g., 10 shots) | Large (e.g., 1/3 total classes)
Small ($\ll$ base, e.g., 3 classes) |

### B.2   DISCUSSION ON LOCAL TESTING AND GLOBAL TESTING

In this section, we will first introduce the pipeline of TPP (Niu et al., 2024b), followed by a comprehensive analysis of the two evaluation paradigms: *local testing* and *global testing*, supported by empirical results.

#### B.2.1   APPROACHES

**TPP**[4]. TPP follows the *local testing* manner and proposes to use a Laplacian smoothing approach to generate a prototypical embedding for each graph task for task ID prediction. As for the $i$-th task with graph data $\mathcal{G}_{s_i} = \mathcal{G}_{q_i} = (\mathbf{A}_{s_i}, \mathbf{X}_{s_i}, \mathcal{V}_{s_i}^l, \mathcal{V}_{s_i}^u)$, TPP constructed a task prototype $\mathbf{p}_{s_i}$ based on the train set $\{\mathbf{x}_j \mid v_j \in \mathcal{V}_{s_i}^l\}$. Specifically, given training graph $\mathcal{G}_{s_i}$, the Laplacian smoothing is first applied to obtain the node embeddings $\mathbf{Z}_{s_i}$:

$$\mathbf{Z}_{s_i} = (\mathbf{I} - (\hat{\mathbf{D}}_{s_i})^{-\frac{1}{2}} \hat{\mathbf{L}}_{s_i} (\hat{\mathbf{D}}_{s_i})^{-\frac{1}{2}})^k \mathbf{X}_{s_i} \tag{3}$$

where $k$ is the smoothing steps, $\mathbf{I}$ is an identity matrix, $\hat{\mathbf{L}}_{s_i}$ is the graph Laplacian matrix of $\hat{\mathbf{A}}_{s_i}$, and $\hat{\mathbf{D}}_{s_i}$ is the diagonal degree matrix. Consequently, the training task prototype $\mathbf{p}_{s_i}$ can be generated as:

$$\mathbf{p}_{s_i} = \frac{1}{|\mathcal{V}_{s_i}^l|} \sum_{v_j \in \mathcal{V}_{s_i}^l} \mathbf{z}_{s_i,j} (\hat{\mathbf{D}}_{s_i,jj})^{-\frac{1}{2}} \tag{4}$$

Thus, all the task prototypes can be separately constructed and stored as $\mathcal{P} = \{\mathbf{p}_{s_1}, \mathbf{p}_{s_2}, ..., \mathbf{p}_{s_n}\}$. Similarly, the task prototype of the local testing on $\mathcal{G}_{q_i} = \mathcal{G}_{s_i}$ can be denoted as:

$$\mathbf{p}_{q_i} = \frac{1}{|\mathcal{V}_{q_i}^u|} \sum_{v_j \in \mathcal{V}_{q_i}^u} \mathbf{z}_{q_i,j} (\hat{\mathbf{D}}_{q_i,jj})^{-\frac{1}{2}} \tag{5}$$

Then, during local testing, the task ID prediction is conducted between the train and test prototypes:

$$t_{q_i} = \arg\min(d(\mathbf{p}_{q_i}, \mathbf{p}_{s_1}), d(\mathbf{p}_{q_i}, \mathbf{p}_{s_2}), ..., d(\mathbf{p}_{q_i}, \mathbf{p}_{s_n})) \tag{6}$$

After precisely identifying the task ID, TPP subsequently incorporates a graph prompt learning framework that assigns distinct prompt vectors to adapt a pretrained GNN for diverse tasks:

$$\overline{\mathbf{x}}_{q_i,j} = \mathbf{x}_{q_i,j} + \sum_m^M \alpha_m \phi_{i,m}, \alpha_m = \frac{e^{(\mathbf{w}_m)^T \mathbf{x}_{q_i,j}}}{\sum_l^M e^{(\mathbf{w}_l)^T \mathbf{x}_{q_i,j}}} \tag{7}$$

---

[4]In this work, we focus solely on formalizing the pipeline of TPP; for rigorous mathematical derivations and proofs, we direct readers to the original literature.

where $\Phi_i = [\phi_{i,1}, \phi_{i,2}, ..., \phi_{i,M}]^T \in \mathbb{R}^{M \times F}$ represents the learnable graph prompt for the $i$-th task, with $M$ denoting the number of vector-based tokens $\phi_i$. $\alpha_m$ is the importance score of token $\phi_{i,m}$ in the prompt, while $\mathbf{w}_j$ corresponds to a learnable projection weight. The final prompted graph representation $\mathcal{G}_{q_i} = \{\mathbf{A}_{q_i}, \mathbf{X}_{q_i} + \Phi_i\}$ is then processed by a frozen pre-trained GNN model $f(\cdot, \cdot)$ to generate predictions:

$$\hat{\mathcal{Y}}_{q_i} = \varphi_i(f(\mathbf{A}_{q_i}, \mathbf{X}_{q_i} + \Phi_i)) \tag{8}$$

where $\varphi_i$ denotes the task-specific MLP layer for the $i$-th task. Crucially, **TPP effectively transforms the class-incremental learning problem into a task-incremental learning paradigm, decomposing the overall continual learning objective into multiple independent sub-tasks, thereby significantly mitigating the learning difficulty**.

**Mean Pooling**. TPP proposes that when applying a sufficiently large number of Laplacian smoothing iterations $k$, the distance between training and test prototypes from the same task approaches zero, while maintaining substantial separation between prototypes belonging to different CIL tasks. This discriminative property ensures reliable prediction accuracy. To evaluate the effectiveness of this technique, we implement an ablation model called Mean Pooling, where the Laplacian smoothing operation is replaced by a basic mean pooling aggregation, formally defined as:

$$\mathbf{p}_{s_i} = \frac{1}{|\mathcal{V}_{s_i}^l|} \sum_{v_j \in \mathcal{V}_{s_i}^l} \mathbf{x}_{s_i,j}, \ \ \mathbf{p}_{q_i} = \frac{1}{|\mathcal{V}_{q_i}^u|} \sum_{v_j \in \mathcal{V}_{q_i}^u} \mathbf{x}_{q_i,j} \tag{9}$$

Under this formulation, the distance between training and test prototypes depends solely on the feature distributions of the training and testing nodes.

**MLP**. By decomposing the CIL task into independent subtasks, TPP eliminates the need for session-wide classification across all accumulated classes. Instead, the model can focus exclusively on individual subtasks, significantly reducing task complexity. Building upon this observation, we propose an additional ablation model employing a basic two-layer MLP for classification:

$$\hat{\mathcal{Y}}_{q_i} = \varphi_i(\mathbf{X}_{q_i}) \tag{10}$$

### B.2.2 EXPERIMENTAL RESULTS

To investigate the impact of Laplacian smoothing and assess subtask complexity, we evaluate the performance of TPP against two variants, Mean Pooling and MLP, with results presented in Table 1. We employ the *local testing* paradigm, where model performance is quantified using two key metrics: Average Accuracy (AA) and Average Forgetting (AF). These metrics are defined as follows:

$$\text{AA} = \frac{1}{n} \sum_{j=1}^{n} (\mathcal{A}_{n,j}), \ \ \text{AF} = \frac{1}{n} \sum_{j=1}^{n-1} (\mathcal{A}_{n,j} - \mathcal{A}_{j,j}) \tag{11}$$

where $n$ denotes the total number of sessions, and $\mathcal{A}_{i,j}$ represents the model's classification accuracy for task $\mathcal{T}_{q_j}$ after training on task $\mathcal{T}_{s_i}$. The experimental results yield two key conclusions:

♣ **Laplacian smoothing demonstrates limited necessity for accurate task identification.** Comparative analysis between TPP and Mean Pooling demonstrates that both approaches achieve accurate task ID prediction. This phenomenon can be explained by the *local testing* setup, where training and testing nodes are drawn from the same graph ($\mathcal{G}_{s_i} = \mathcal{G}_{q_i}$). In this configuration, the feature distribution similarity between training and testing nodes within each class provides sufficient discriminative information for reliable task identification. These results indicate that the *local testing* paradigm is **inherently susceptible to task ID leakage**.

♠ **Decomposition into subtasks significantly reduces task complexity.** As evidenced by the experimental results, the MLP baseline achieves comparable performance to TPP across Citeseer, WikiCS, Arxiv-23, and Arxiv datasets, with merely a 3.08% average performance gap. This finding demonstrates that task decomposition effectively reduces learning difficulty, enabling simpler architectures to attain competitive performance. However, this characteristic **may compromise the evaluation of models' true continual learning capabilities**, as the simplified subtasks fail to fully capture the challenges inherent in the original CIL problem.

Table 6: Statistics and experimental settings for all datasets in LLM4GCL. The symbol '*' indicates adjusted statistics that differ from the original dataset.

| | Dataset | Cora | Citeseer | WikiCS | Photo | Products | Arxiv-23 | Arxiv |
|---|---|---|---|---|---|---|---|---|
| **Attributes** | # Nodes | 2,708 | 3,186 | 11,701 | 48,362 | 53,994* | 46,196* | 169,343 |
| | # Edges | 5,429 | 4,277 | 215,863 | 500,928 | 72,242* | 78,542* | 1,166,243 |
| | # Features | 1,433 | 3,703 | 300 | 768 | 100 | 300 | 128 |
| | # Classes | 7 | 6 | 10 | 12 | 31* | 37* | 40 |
| | Avg. # Token | 183.4 | 210.0 | 629.9 | 201.5 | 152.9* | 237.7* | 239.8 |
| | Domain | Citation | Citation | Web link | E-Commerce | E-Commerce | Citation | Citation |
| **NCIL** | # Classes per session | 2 | 2 | 3 | 3 | 4 | 4 | 4 |
| | # Sessions | 3 | 3 | 3 | 4 | 8 | 9 | 10 |
| | # Shots per class | 100 | 100 | 200 | 400 | 400 | 400 | 800 |
| **FSNCIL** | # Base Classes | 3 | 2 | 4 | 4 | 11 | 13 | 12 |
| | # Novel Classes | 4 | 4 | 6 | 8 | 20 | 24 | 28 |
| | # Ways per session | 2 | 2 | 3 | 4 | 4 | 4 | 4 |
| | # Sessions | 3 | 3 | 3 | 3 | 6 | 7 | 8 |
| | # Shots per base class | 100 | 100 | 200 | 400 | 400 | 400 | 800 |
| | # Shots per novel class | 5 | 5 | 5 | 5 | 5 | 5 | 5 |

## B.3 DETAILS OF DATASETS

All of the public datasets used in `LLM4GCL` were previously published, covering a multitude of domains. For each dataset, we store the graph-type data in the .pt format using PyTorch. This includes shallow embeddings, raw text of nodes, edge indices, node labels, and label names. The statistics of these datasets are presented in Table 6, and the detailed descriptions are listed in the following:

- **Cora** (Chen et al., 2024b). The Cora dataset is a citation network comprising research papers and their citation relationships within the computer science domain. The raw text data for the Cora dataset was sourced from the GitHub repository provided in Chen et al.[5]. In this dataset, each node represents a research paper, and the raw text feature associated with each node includes the title and abstract of the respective paper. An edge in the Cora dataset signifies a citation relationship between two papers. The label assigned to each node corresponds to the category of the paper.

- **Citeseer** (Chen et al., 2024b). The Citeseer dataset is a citation network comprising research papers and their citation relationships within the computer science domain. The TAG version of the dataset contains text attributes for 3,186 nodes, and the raw text data for the Citeseer dataset was sourced from the GitHub repository provided in Chen et al.. Each node represents a research paper, and each edge signifies a citation relationship between two papers.

- **WikiCS** (Liu et al., 2023a). The WikiCS dataset is an internet link network where each node represents a Wikipedia page, and each edge represents a reference link between pages. The raw text data for the WikiCS dataset was collected from OFA[6]. The raw text associated with each node includes the name and content of a Wikipedia entry. Each node's label corresponds to the category of the entry.

- **Ele-Photo (Photo)** (Yan et al., 2023). The Ele-Photo dataset[7] is derived from the Amazon Electronics dataset (Ni et al., 2019). The nodes in the dataset represent electronics-related products, and edges between two products indicate frequent co-purchases or co-views. The label for each dataset corresponds to the three-level label of the electronics products. User reviews on the item serve as its text attribute. In cases where items have multiple reviews, the review with the highest number of votes is utilized. For items lacking highly-voted reviews, a user review is randomly chosen as the text attribute.

- **Ogbn-products (Products)** (He et al., 2023). The OGBN-Products dataset is characterized by its large scale, originally containing approximately 2 million nodes and 61 million edges. Following the node sampling strategy proposed in TAPE[8], we utilize a subset consisting of about 54k nodes

---

[5] https://github.com/CurryTang/Graph-LLM
[6] https://github.com/LechengKong/OneForAll
[7] https://github.com/sktsherlock/TAG-Benchmark
[8] https://github.com/XiaoxinHe/TAPE

and 72k edges for experimental use. In this dataset, each node represents a product sold on Amazon, and edges between two products indicate frequent co-purchasing relationships. The raw text data for each node includes product titles, descriptions, and/or reviews.

- **Arxiv-23** (He et al., 2023). The Arxiv-23 dataset is proposed in TAPE, representing a directed citation network comprising computer science arXiv papers published in 2023 or later. Similar to Ogbn-arxiv, each node in this dataset corresponds to an arXiv paper, and directed edges indicate citation relationships between papers. The raw text data for each node includes the title and abstract of the respective paper. The label assigned to each node represents one of the 40 subject areas of arXiv CS papers, such as cs.AI, cs.LG, and cs.OS. These subject areas are manually annotated by the paper's authors and arXiv moderators.

- **Ogbn-arxiv (Arxiv)** (Liu et al., 2023a). The Ogbn-arxiv dataset is a citation network comprising papers and their citation relationships, collected from the arXiv platform. The raw text data for the dataset was sourced from the GitHub repository provided in OFA. The original raw texts are available here[9]. Each node represents a paper, and each edge denotes a citation relationship.

## B.4 DETAILS OF BASELINE MODELS

In LLM4GCL, we integrate 15 baseline methods for NCIL and FSNCIL tasks, including 6 GNN-based methods, 4 LLM-based methods, and 5 GLM-based methods. We provide detailed descriptions of these methods used in our benchmark as follows.

- **GNN-based methods**.
  - **GCN** (Kipf & Welling, 2016). The first deep learning model that leverages graph convolutional layers. In LLM4GCL, we employ a two-layer GCN followed by one MLP layer. We use the code available at `https://github.com/pyg-team/pytorch_geometric`.
  - **EWC** (Kirkpatrick et al., 2017). A regularization-based CL method that prevents catastrophic forgetting through quadratic penalties on key parameter deviations. Parameter importance is quantified via the Fisher information matrix, maintaining performance on learned tasks during new task acquisition. We use the code available at `https://github.com/QueuQ/CGLB`.
  - **LwF** (Li & Hoiem, 2017). LwF minimizes the discrepancy between the logits of the old model and the new model through knowledge distillation to preserve knowledge from the old tasks. We use the code available at `https://github.com/QueuQ/CGLB`.
  - **Cosine**. Cosine first trains a GNN on the initial task session, subsequently freezes the network parameters, and leverages a training-free prototype mechanism to generate task-specific prototypes, ultimately employing cosine similarity for classification.
  - **TPP** (Niu et al., 2024b). A replay-and-forget-free GCL approach that employs Laplacian smoothing-based task identification to precisely predict task IDs and incorporates graph prompt engineering to adaptively transform the GNN into a series of task-specific classification modules. We use the code available at `https://github.com/mala-lab/TPP`.
  - **TEEN** (Wang et al., 2023a). TEEN is a training-free prototype calibration strategy that addresses the issue of new classes being misclassified into base classes by leveraging the semantic similarity between base and new classes, which is captured by a feature extractor pre-trained on base classes. We use the code available at `https://github.com/wangkiw/TEEN`.

- **LLM-based methods**.
  - **BERT** (Bhargava et al., 2021; Devlin et al., 2019). A Transformer-based model family that employs masked language modeling and next sentence prediction pretraining to learn bidirectional contextual representations. It achieves state-of-the-art performance across multiple NLP tasks with minimal task-specific modifications. We use the code available at `https://github.com/google-research/bert` and `https://github.com/prajjwal1/generalize_lm_nli`.
  - **RoBERTa** (Liu et al., 2019). An optimized BERT variant family that enhances pretraining through dynamic masking, larger batches, and extended data. It removes the next sentence prediction objective while maintaining MLM, achieving superior performance across NLP benchmarks with more efficient training. We use the code available at `https://github.com/facebookresearch/fairseq/tree/main/examples/xlmr`.

---

[9]`https://snap.stanford.edu/ogb/data/misc/ogbn_arxiv`

- **LLaMA** (Touvron et al., 2023). A foundational large language model family featuring architectural optimizations like RMSNorm and SwiGLU activations. Trained on trillions of tokens from diverse public corpora, it achieves remarkable few-shot performance while maintaining efficient inference compared to similarly-sized models. We use the code available at `https://github.com/meta-llama/llama-cookbook`.
  - **SimpleCIL** (Zhou et al., 2025). A computationally efficient approach that freezes the PTM's embedding function and derives class prototypes by averaging embeddings for classification, which harnesses the model's inherent generalizability for effective knowledge transfer without fine-tuning. We use the code available at `https://github.com/LAMDA-CL/RevisitingCIL`.

- **GLM-based methods**.
  - **GCN$_{\text{Emb}}$**. A simple method enhances GNNs by replacing their shallow embeddings with deep LLM-generated embeddings, where node text descriptions are encoded using LLaMA-8B to improve representation quality.
  - **ENGINE** (Zhu et al., 2024). A framework adds a lightweight and tunable G-Ladder module to each layer of the LLM, which uses a message-passing mechanism to integrate structural information. This enables the output of each LLM layer (i.e., token-level representations) to be passed to the corresponding G-Ladder, where the node representations are enhanced and then used for node classification. We use the code available at `https://github.com/ZhuYun97/ENGINE`.
  - **GraphPropmter** (Liu et al., 2024). A framework integrating graph structures and LLMs via adaptive prompts, projecting GNN-based structural embeddings through MLP to align with the LLM's semantic space for joint topology-text processing. We use the code available at `https://github.com/franciscoliu/graphprompter`.
  - **GraphGPT** (Tang et al., 2024a). A framework that initially aligns the graph encoder with natural language semantics through text-graph grounding, and then combines the trained graph encoder with the LLM using a projector, through which the model can directly complete graph tasks with natural language, thus performing zero-shot transferability. We use the code available at `https://github.com/WxxShirley/LLMNodeBed`.
  - **LLaGA** (Chen et al., 2024a). LLaGA uses node-level templates to transform graph data into structured sequences, then maps them into token embeddings, enabling LLMs to process graph data with improved generalizability. We use the code available at `https://github.com/WxxShirley/LLMNodeBed`.

## C  DETAILS ON IMPLEMENTATIONS

### C.1  HYPER-PARAMETERS

- For GCN, we grid-search the hyperparameters

```
lr in [1e-5, 1e-4, 1e-3], hidden_dim in [64, 128, 256]
layer_num = 2, dropout = 0.5
```

- For EWC, we grid-search the hyperparameters

```
lr in [1e-5, 1e-4, 1e-3], hidden_dim in [64, 128, 256],
strength in [1, 100, 10000], layer_num = 2
dropout = 0.5
```

- For LwF, we grid-search the hyperparameters

```
lr in [1e-5, 1e-4, 1e-3], hidden_dim in [64, 128, 256],
strength in [1, 100, 10000], lambda in [0.1, 1.0]
T in [0.2, 2.0], layer_num = 2, dropout = 0.5
```

- For Cosine, we grid-search the hyperparameters

```
            lr in [1e-5, 1e-4, 1e-3], hidden_dim in [64, 128, 256]
            layer_num = 2, dropout = 0.5, T = 1.0, sample_num = 100
```

- For TPP, we grid-search the hyperparameters

```
            lr in [1e-5, 1e-4, 1e-3], hidden_dim in [64, 128, 256],
            pe in [0.2, 0.3], pf in [0.2, 0.3], layer_num = 2
            dropout = 0.5, pretrain_batch_size = 500
            pretrain_lr = 1e-3
```

- For TEEN, we grid-search the hyperparameters

```
            lr in [1e-5, 1e-4, 1e-3], hidden_dim in [64, 128, 256]
            layer_num = 2, dropout = 0.5, T = 1.0, sample_num = 100
            softmax_T = 16, shift_weight = 0.5
```

- For BERT and RoBERTa, we grid-search the hyperparameters

```
            lr in [1e-4, 2e-4, 5e-4], batch_size in [5, 10, 20]
            min_lr = 5e-6, weight_decay = 5e-2, dropout = 0.1
            att_drouput = 0.1, max_length = 256, lora_r = 5
            lora_alpha = 16, lora_dropout = 0.05
```

- For LLaMA, we grid-search the hyperparameters

```
            lr in [1e-4, 2e-4, 5e-4], batch_size in [5, 10, 20]
            min_lr = 5e-6, weight_decay = 5e-2, dropout = 0.1
            att_drouput = 0.1, max_length = 512, lora_r = 5
            lora_alpha = 16, lora_dropout = 0.05
```

- For SimpleCIL, we grid-search the hyperparameters

```
            lr in [1e-4, 2e-4, 5e-4], batch_size in [5, 10, 20]
            min_lr = 5e-6, weight_decay = 5e-2, dropout = 0.1
            att_drouput = 0.1, max_length = 256, lora_r = 5
            lora_alpha = 16, lora_dropout = 0.05, T = 1.0
            sample_num = 20
```

- For $GCN_{Emb}$, we grid-search the hyperparameters

```
            lr in [1e-5, 1e-4, 1e-3], hidden_dim in [64, 128, 256]
            layer_num = 2, dropout = 0.5
```

- For ENGINE, we grid-search the hyperparameters

```
            lr in [1e-5, 1e-4, 1e-3], hidden_dim in [64, 128, 256]
            r in [1, 32], layer_num = 1, dropout = 0.5, T = 0.1
            layer_select = [0, 6, 12, 18, 24, -1]
```

- For GraphPrompter, we follow the instructions from the paper (Liu et al., 2024), use

```
            lr = 2e-4, batch_size = 10, min_lr = 5e-6
            weight_decay = 5e-2, dropout = 0.1, att_drouput = 0.1
            max_length = 512, lora_r = 5, lora_alpha = 16
            lora_dropout = 0.05, gnn = 'GCN', layer_num = 4
            hidden_dim = 1024, output_dim = 1024,
            dropout = 0.5, proj_hidden_dim = 1024
```

Table 7: Selection of GNN and LLM Backbones in `LLM4GCL`, $\mathbf{A}, \mathbf{X}, \mathcal{R}$ refers to the adjacent matrix, feature matrix, and raw text attributes of the input TAG, respectively.

| Type | Method | Predictor | Input | Output Format | GNN Backbone | LLM Backbone |
|------|--------|-----------|-------|---------------|--------------|--------------|
| GNN-based | GCN Kipf & Welling (2016) | GNN | $(\mathbf{A}, \mathbf{X})$ | Logits | GCN | - |
| | EWC Kirkpatrick et al. (2017) | GNN | $(\mathbf{A}, \mathbf{X})$ | Logits | GCN | - |
| | LwF Li & Hoiem (2017) | GNN | $(\mathbf{A}, \mathbf{X})$ | Logits | GCN | - |
| | Cosine | GNN | $(\mathbf{A}, \mathbf{X})$ | Logits | GCN | - |
| | TPP Niu et al. (2024b) | GNN | $(\mathbf{A}, \mathbf{X})$ | Logits | SGC, GCN | - |
| | TEEN Wang et al. (2023a) | GNN | $(\mathbf{A}, \mathbf{X})$ | Logits | GCN | - |
| LLM-based | BERT Devlin et al. (2019) | LLM | $\mathcal{R}$ | Logits | - | BERT-base (110M) |
| | RoBERTa Liu et al. (2019) | LLM | $\mathcal{R}$ | Logits | - | RoBERTa-large (355M) |
| | LLaMA Touvron et al. (2023) | LLM | $\mathcal{R}$ | Prediction Texts | - | LLaMA3-8B |
| | SimpleCIL Zhou et al. (2025) | LLM | $\mathcal{R}$ | Logits | - | RoBERTa-large (355M) |
| GLM-based | GCN$_{\text{Emb}}$ | GNN | $(\mathbf{A}, \mathcal{R})$ | Logits | GCN | LLaMA3-8B |
| | ENGINE Zhu et al. (2024) | GNN | $(\mathbf{A}, \mathcal{R})$ | Logits | GCN | LLaMA3-8B |
| | GraphPrompter Huang et al. (2024) | LLM | $(\mathbf{A}, \mathbf{X}, \mathcal{R})$ | Prediction Texts | GCN | LLaMA3-8B |
| | GraphGPT Tang et al. (2024a) | LLM | $(\mathbf{A}, \mathbf{X}, \mathcal{R})$ | Prediction Texts | - | LLaMA3-8B |
| | LLaGA Chen et al. (2024a) | LLM | $(\mathbf{A}, \mathbf{X}, \mathcal{R})$ | Prediction Texts | - | LLaMA3-8B |

- For GraphGPT, we follow the instructions from the paper (Wu et al., 2025), use

```
dropout = 0.1, att_drouput = 0.1, max_length = 512
s1_k_hop = 2, s1_num_neighbors = 5
s1_max_txt_length = 512, s1_epoch = 2, s1_batch_size = 5
s1_lr = 1e-4, s2_num_neighbors = 4, max_txt_length = 512
s2_epoch = 10, s2_batch_size = 5, s2_lr = 1e-4
lora_r = 5, lora_alpha = 16, lora_dropout = 0.05
```

- For LLaGA, we follow the instructions from the paper (Chen et al., 2024a), use

```
lr = 2e-4, batch_size = 10, min_lr = 5e-6
weight_decay = 5e-2, dropout = 0.1, att_drouput = 0.1
max_length = 512, llm_freeze = 'True'
neighbor_template = 'ND', nd_mean = True, k_hop = 2
sample_size = 10, hop_field = 4, proj_layer = 2
```

- For SimGCL, we follow the instructions from the paper (Wang et al., 2025), use

```
lr = 2e-4, batch_size = 10, min_lr = 5e-6
weight_decay = 5e-2, lora_r = 5, lora_alpha = 16
lora_dropout = 0.05, T = 1.0, sample_num = 50
hop = [20, 20], include_label = False, max_node_text_len = 128
```

## C.2 BACKBONE SELECTION

To compare the various baseline methods in `LLM4GCL` as fairly as possible, we try to utilize the same GNN and LLM backbones in our implementations. Table 7 shows the GNN and LLM backbones used in the original implementations, as well as those implemented in `LLM4GCL`.

## D TEXT PROMPT DESIGN

For GraphPrompter (Liu et al., 2024), GraphGPT (Tang et al., 2024a), and LLaGA (Chen et al., 2024a), we utilize the prompt templates provided in their original papers for several datasets, including Cora and arXiv. For datasets not originally addressed, such as Photo, we adapt their prompt designs to create similarly formatted prompts. For SimGCL, we adopt the graph prompt template established by Wang et al. (Wang et al., 2025). Below is a summary of these prompt templates using Cora as an example, where ⟨**labels**⟩ denotes the dataset-specific label space, ⟨**graph**⟩ represents the tokenized graph context, ⟨**raw_text**⟩ and ⟨**paper_titles**⟩ refer to the node's original raw text, and ⟨**target**

node⟩ is the node's id within the graph. It is important to emphasize that during the continual learning process, as new classes emerge sequentially, the content of ⟨**labels**⟩ dynamically expands to incorporate these newly introduced class labels.

---

**Illustration of Prompts Utilized by LLaMA and GraphPrompter on the Cora Dataset**

**Cora:** ⟨**raw_text**⟩. Which of the following sub-categories of AI does this paper belong to? Here are the |⟨**labels**⟩| categories: ⟨**labels**⟩. Reply with only one category that you think this paper might belong to. Only reply to the category phrase without any other explanatory words.

---

**Illustration of Prompts Utilized by GraphGPT on the Cora Dataset**

**Matching:** Given a sequence of graph tokens ⟨**graph**⟩ that constitute a subgraph of a citation graph, where the first token represents the central node of the subgraph, and the remaining nodes represent the first or second-order neighbors of the central node. Each graph token contains the title and abstract information of the paper at this node. Here is a list of paper titles: ⟨**paper_titles**⟩. Please reorder the list of papers according to the order of graph tokens (i.e., complete the matching of graph tokens and papers).

**Instruction:** Given a citation graph: ⟨**graph**⟩ where the 0th node is the target paper, with the following information: ⟨**raw_text**⟩. Question: Which of the following specific research does this paper belong to: ⟨**labels**⟩. Directly give the full name of the most likely category of this paper.

---

**Illustration of Prompts Utilized by LLaGA on the Cora Dataset**

**System:** You are a helpful language and graph assistant. You can understand the graph content provided by the user and assist with the node classification task by outputting the most likely label.

**Instruction:** Given a node-centered graph: ⟨**graph**⟩, each node represents a paper, we need to classify the center node into |⟨**labels**⟩| classes: ⟨**labels**⟩, please tell me which class the center node belongs to?

---

**Illustration of Prompts Utilized by SimGCL on the Cora Dataset**

**System:** You are a good graph reasoner. Given a graph description from the Cora dataset, understand the structure and answer the question.

**Instruction:** ⟨**target node**⟩ ⟨**raw_text**⟩, known neighbor papers at hop 1: ⟨**1-hop neighbors**⟩ ⟨**raw_texts**⟩, known neighbor papers at hop 2: ⟨**2-hop neighbors**⟩ ⟨**raw_texts**⟩. **Question**: Please predict which of the following sub-categories of AI this paper belongs to. Choose from the following categories: ⟨**labels**⟩.

---

# E    ADDITIONAL EXPERIMENTAL RESULTS

## E.1    SESSION NUMBERS

Table 8 presents the performance of baseline models and SimGCL on the Arxiv dataset under different session-class configurations. Consistent with **Obs. ❽**, as session numbers increase, the performance advantages of BERT, GraphGPT, and LLaGA over GNN-based approaches diminish significantly, demonstrating pronounced catastrophic forgetting in long-session scenarios.

## E.2    EXTENDED ANALYSIS OF SIMGCL

To comprehensively assess SimGCL's performance, we conducted an evaluation using 1-hop neighborhood aggregation, with detailed results presented in Table 9 and Table 10.

**Observation D.❶ Rich structural information better enhances task comprehension**. In Table 9 and Table 10, SimGCL exhibits performance degradation when limited to 1-hop neighborhood information, particularly on densely-connected datasets, WikiCS and Photo, where it underperforms even the vanilla GCN baseline. We hypothesize this stems from an inadequate structural context, misleading the model's representation learning. Notably, modest improvements occur on sparser graph datasets, Cora and Citeseer, we attribute this to their smaller scale and lower connectivity makes 1-hop information sufficient for effective prediction.

### E.3 DOMAIN-INCREMENTAL LEARNING SETTING

To extend the evaluation scope of the proposed `LLM4GCL` framework to the domain-incremental learning (DIL) scenario, we conducted comprehensive experiments comparing various baseline methods with SimGCL under domain-sequential learning conditions. Specifically, each model was trained incrementally on the PubMed, CiteSeer, and Cora datasets in sequence. The experimental results are presented in Table 11, yielding the following key observations:

**Observation D.❷. LLM- and GLM-based methods demonstrate stronger domain-incremental learning capabilities compared to GNN-based approaches.** In the domain-incremental learning (DIL) setting, most LLM-based and GLM-based methods consistently outperform training-from-scratch GNNs. This underscores the strong generalization ability of pre-trained LLMs and highlights their potential in addressing DIL challenges within graph-based learning. However, we note that the performance margin of GLM-based models is narrower than that of LLaMA and SimpleCIL. We attribute this limitation to the architectural discrepancy between shallow GNNs and deep LLMs (e.g., GraphPrompter), as well as the misalignment between textual and graph structural representations (e.g., GraphGPT), which collectively impair the generative capability of these models.

**Observation D.❸ SimpleCIL and SimGCL maintain substantial performance advantages over other baselines in the DIL setting.** Experimental results show that SimpleCIL and SimGCL reach the two highest performances, exhibiting significantly less forgetting and greater performance gain compared to GCN, in contrast with all other baselines. These outcomes confirm that class prototypes effectively preserve the generalization capabilities of LLMs across diverse domains, while graph structural prompts enhance their ability to comprehend and adapt to unseen graph topologies.

### E.4 TIME EFFICIENCY

To more comprehensively assess the computational overhead associated with LLMs and GLMs and to better analyze the trade-off between time efficiency and performance, we compared the computational costs of SimGCL against all baseline methods on the Cora dataset. Results are provided in Table 12.

Table 8: Performance comparison of `LLM4GCL` baselines in NCIL scenario across different class sizes (W) and session numbers (S) on Arxiv. The best and second results are highlighted.

| Methods | 8W5S | | 5W8S | | 4W10S | | 2W20S | |
|---|---|---|---|---|---|---|---|---|
| | $\bar{\mathcal{A}}$ | $\mathcal{A}_N$ | $\bar{\mathcal{A}}$ | $\mathcal{A}_N$ | $\bar{\mathcal{A}}$ | $\mathcal{A}_N$ | $\bar{\mathcal{A}}$ | $\mathcal{A}_N$ |
| GCN | 24.5 ↓0.0 | 2.0 ↓0.0 | 20.5 ↓0.0 | 1.1 ↓0.0 | 21.2 ↓0.0 | 1.5 ↓0.0 | 13.4 ↓0.0 | 0.7 ↓0.0 |
| EWC | 31.6 ↑7.1 | 19.6 ↑17.6 | 26.2 ↑5.7 | 13.8 ↑12.7 | 24.9 ↑3.7 | 13.8 ↑12.3 | 15.4 ↑2.0 | 4.2 ↑3.5 |
| LwF | 29.2 ↑4.7 | 8.6 ↑6.6 | 22.6 ↑2.1 | 2.7 ↑1.6 | 18.9 ↓2.3 | 1.6 ↑0.1 | 14.2 ↑0.8 | 1.0 ↑0.3 |
| Cosine | 35.6 ↑11.1 | 23.2 ↑21.2 | 38.3 ↑17.8 | 27.9 ↑26.8 | 37.4 ↑16.2 | 27.8 ↑26.3 | 41.6 ↑28.2 | 31.4 ↑30.7 |
| BERT | 36.7 ↑12.2 | 15.0 ↑13.0 | 29.7 ↑9.2 | 10.2 ↑9.1 | 26.0 ↑4.8 | 9.7 ↑8.2 | 16.6 ↑3.2 | 5.0 ↑4.3 |
| RoBERTa | 33.9 ↑9.4 | 3.9 ↑1.9 | 26.7 ↑6.2 | 2.1 ↑1.0 | 24.4 ↑3.2 | 1.4 ↓0.1 | 15.6 ↑2.2 | 0.3 ↓0.4 |
| LLaMA | 35.0 ↑10.5 | 3.8 ↑1.8 | 27.6 ↑7.1 | 1.5 ↑0.4 | 24.9 ↑3.7 | 1.4 ↓0.1 | 15.9 ↑2.5 | 0.3 ↓0.4 |
| SimpleCIL | 46.1 ↑21.6 | 31.4 ↑29.4 | 49.7 ↑29.2 | 35.9 ↑34.8 | 50.6 ↑29.4 | 36.5 ↑35.0 | 52.6 ↑39.2 | 39.1 ↑38.4 |
| GCN_LLMEmb | 33.7 ↑9.2 | 3.8 ↑1.8 | 26.6 ↑6.1 | 2.0 ↑0.9 | 24.3 ↑3.1 | 1.4 ↓0.1 | 15.6 ↑2.2 | 0.3 ↓0.4 |
| ENGINE | 34.1 ↑9.6 | 3.7 ↑1.7 | 26.5 ↑6.0 | 2.0 ↑0.9 | 24.6 ↑3.4 | 1.3 ↓0.2 | 15.7 ↑2.3 | 0.2 ↓0.5 |
| GraphPrompter | 35.1 ↑10.6 | 2.9 ↑0.9 | 27.8 ↑7.3 | 2.0 ↑0.9 | 24.8 ↑3.6 | 1.4 ↓0.1 | 16.8 ↑3.4 | 0.4 ↓0.3 |
| GraphGPT | 38.9 ↑14.4 | 7.4 ↑5.4 | 35.0 ↑14.5 | 6.2 ↑5.1 | 32.2 ↑11.0 | 3.9 ↑2.4 | 23.5 ↑10.1 | 3.1 ↑2.4 |
| LLaGA | 34.8 ↑10.3 | 8.6 ↑6.6 | 27.9 ↑7.4 | 5.3 ↑4.2 | 26.1 ↑4.9 | 1.3 ↓0.2 | 16.6 ↑3.2 | 0.9 ↑0.2 |
| SimGCL (Ours) | 51.6 ↑27.1 | 28.7 ↑26.7 | 53.0 ↑32.5 | 30.4 ↑29.3 | 59.9 ↑38.7 | 33.8 ↑32.3 | 57.4 ↑44.0 | 17.5 ↑16.8 |

Table 9: Performance comparison of GNN-, LLM-, and GLM-based methods in the NCIL scenario of `LLM4GCL`. The best and second results are highlighted.

| Methods | Cora $\bar{A}$ | $\mathcal{A}_N$ | Citeseer $\bar{A}$ | $\mathcal{A}_N$ | WikiCS $\bar{A}$ | $\mathcal{A}_N$ | Photo $\bar{A}$ | $\mathcal{A}_N$ | Products $\bar{A}$ | $\mathcal{A}_N$ | Arxiv-23 $\bar{A}$ | $\mathcal{A}_N$ | Arxiv $\bar{A}$ | $\mathcal{A}_N$ |
|---|---|---|---|---|---|---|---|---|---|---|---|---|---|---|
| GCN | 57.0 | 38.2 | 52.4 | 30.2 | 54.9 | 34.9 | 46.5 | 19.9 | 25.5 | 5.4 | 19.9 | 2.9 | 21.2 | 1.5 |
| EWC | 56.0 | 31.0 | 45.9 | 28.5 | 55.3 | 33.0 | 46.9 | 20.5 | 29.4 | 15.9 | 25.7 | 15.0 | 24.9 | 13.8 |
| LwF | 55.7 | 30.8 | 47.8 | 28.2 | 55.0 | 34.0 | 45.8 | 20.4 | 26.0 | 7.5 | 21.1 | 6.8 | 18.9 | 1.6 |
| Cosine | 65.4 | 45.2 | 50.7 | 31.0 | 66.5 | 53.5 | 63.6 | 49.6 | 36.1 | 16.1 | 36.1 | 24.2 | 37.4 | 27.8 |
| TPP | 45.7 | 13.7 | 45.4 | 9.6 | 35.1 | 9.8 | 36.3 | 5.7 | 15.0 | 0.0 | 12.2 | 0.3 | 19.6 | 3.4 |
| BERT | 56.0 | 29.9 | 53.8 | 28.7 | 58.4 | 30.0 | 43.4 | 18.4 | 26.9 | 4.1 | 27.1 | 5.1 | 26.0 | 9.7 |
| RoBERTa | 54.6 | 29.6 | 54.1 | 28.6 | 55.1 | 30.8 | 43.5 | 19.1 | 27.6 | 3.2 | 23.1 | 0.8 | 24.4 | 1.4 |
| LLaMA | 65.6 | 53.8 | 55.7 | 31.7 | 55.5 | 30.9 | 44.6 | 19.1 | 29.8 | 0.4 | 24.3 | 1.0 | 24.9 | 1.4 |
| SimpleCIL | 70.8 | 58.3 | 66.4 | 49.5 | 71.4 | 57.3 | 62.1 | 52.5 | 66.8 | 52.6 | 52.4 | 38.8 | 50.6 | 36.5 |
| GCN$_{\text{LLMEmb}}$ | 59.1 | 31.1 | 53.6 | 30.4 | 53.4 | 27.5 | 47.7 | 21.0 | 26.9 | 0.1 | 22.9 | 0.8 | 24.3 | 1.4 |
| ENGINE | 59.2 | 31.3 | 53.5 | 29.8 | 56.4 | 30.1 | 47.9 | 21.0 | 27.2 | 1.1 | 22.5 | 0.7 | 24.6 | 1.3 |
| GraphPrompter | 61.9 | 46.8 | 60.2 | 30.6 | 59.6 | 38.3 | 51.5 | 31.0 | 29.0 | 0.8 | 23.4 | 0.9 | 24.8 | 1.4 |
| GraphGPT | 55.5 | 31.6 | 60.0 | 30.1 | 62.0 | 49.2 | 50.8 | 30.2 | 35.5 | 3.2 | 30.7 | 5.8 | 32.2 | 3.9 |
| LLaGA | 58.2 | 30.2 | 51.3 | 27.8 | 53.7 | 27.6 | 47.2 | 20.7 | 25.7 | 0.2 | 26.9 | 4.1 | 26.1 | 1.3 |
| SimGCL-1 Hops | 89.4 | 83.9 | 76.4 | 64.2 | 32.4 | 20.3 | 44.7 | 33.5 | 70.9 | 43.7 | 28.2 | 6.1 | 38.2 | 10.7 |
| SimGCL-2 Hops | 84.6 | 80.0 | 77.1 | 66.3 | 73.5 | 61.9 | 82.1 | 72.6 | 71.1 | 60.2 | 38.7 | 13.6 | 59.9 | 33.8 |

Table 10: Performance comparison of GNN-, LLM-, and GLM-based methods in the FSNCIL scenario of `LLM4GCL`. The best and second results are highlighted.

| Methods | Cora $\bar{A}$ | $\mathcal{A}_N$ | Citeseer $\bar{A}$ | $\mathcal{A}_N$ | WikiCS $\bar{A}$ | $\mathcal{A}_N$ | Photo $\bar{A}$ | $\mathcal{A}_N$ | Products $\bar{A}$ | $\mathcal{A}_N$ | Arxiv-23 $\bar{A}$ | $\mathcal{A}_N$ | Arxiv $\bar{A}$ | $\mathcal{A}_N$ |
|---|---|---|---|---|---|---|---|---|---|---|---|---|---|---|
| GCN | 68.0 | 38.1 | 39.5 | 17.4 | 62.4 | 47.4 | 58.5 | 32.3 | 36.0 | 14.1 | 22.2 | 5.4 | 25.3 | 2.0 |
| EWC | 59.0 | 36.3 | 49.2 | 21.2 | 58.4 | 40.4 | 62.0 | 28.4 | 45.7 | 31.5 | 36.4 | 29.5 | 31.5 | 18.3 |
| LwF | 63.3 | 43.5 | 45.1 | 20.7 | 59.4 | 41.0 | 60.1 | 29.4 | 50.3 | 38.7 | 30.8 | 15.1 | 27.8 | 3.8 |
| Cosine | 72.6 | 57.8 | 49.1 | 25.7 | 68.0 | 50.7 | 67.9 | 50.5 | 50.9 | 33.6 | 40.1 | 27.9 | 27.2 | 17.2 |
| TEEN | 60.9 | 40.3 | 59.0 | 39.5 | 59.3 | 42.4 | 59.3 | 35.5 | 49.6 | 28.4 | 39.2 | 27.1 | 31.8 | 18.6 |
| TPP | 39.0 | 9.1 | 37.3 | 12.6 | 37.3 | 12.4 | 40.0 | 14.0 | 14.0 | 0.0 | 7.3 | 0.0 | 11.1 | 6.0 |
| BERT | 56.4 | 34.7 | 61.1 | 38.8 | 61.7 | 33.0 | 47.7 | 25.8 | 22.4 | 3.8 | 15.2 | 3.4 | 14.9 | 6.0 |
| RoBERTa | 59.6 | 41.9 | 54.0 | 29.2 | 67.2 | 42.3 | 58.2 | 29.6 | 38.8 | 6.9 | 22.6 | 1.4 | 25.7 | 1.3 |
| LLaMA | 72.6 | 55.6 | 75.9 | 55.5 | 65.2 | 43.9 | 61.4 | 32.3 | 43.5 | 13.7 | 23.5 | 1.5 | 22.7 | 1.4 |
| SimpleCIL | 69.6 | 53.6 | 64.1 | 49.9 | 73.2 | 63.1 | 66.3 | 53.0 | 65.6 | 53.6 | 49.8 | 40.0 | 46.4 | 36.6 |
| GCN$_{\text{LLMEmb}}$ | 68.2 | 40.1 | 54.3 | 28.7 | 54.7 | 31.0 | 66.0 | 34.2 | 30.6 | 0.2 | 21.4 | 1.0 | 22.1 | 0.9 |
| ENGINE | 52.2 | 28.3 | 47.6 | 25.5 | 46.8 | 23.7 | 48.0 | 21.5 | 20.9 | 0.1 | 17.4 | 0.5 | 17.3 | 1.4 |
| GraphPrompter | 63.2 | 37.3 | 65.3 | 34.5 | 69.5 | 51.6 | 69.7 | 51.0 | 37.9 | 2.2 | 27.4 | 2.3 | 27.3 | 1.0 |
| GraphGPT | 62.4 | 39.6 | 65.0 | 41.7 | 71.2 | 61.6 | 62.2 | 38.9 | 43.2 | 16.2 | 25.4 | 1.7 | 24.3 | 2.0 |
| LLaGA | 62.0 | 39.6 | 52.2 | 28.0 | 49.4 | 30.4 | 48.8 | 18.0 | 28.0 | 0.2 | 18.2 | 0.9 | 16.6 | 1.6 |
| SimGCL-1 Hop | 85.5 | 77.9 | 77.3 | 63.5 | 30.4 | 23.2 | 50.6 | 44.8 | 57.2 | 30.1 | 19.9 | 4.2 | 25.9 | 3.9 |
| SimGCL-2 Hops | 78.0 | 67.6 | 78.0 | 63.8 | 73.5 | 61.9 | 81.2 | 71.3 | 69.7 | 62.7 | 38.7 | 13.6 | 59.9 | 33.8 |

The results demonstrate that SimGCL is more efficient than other LLaMA-based methods (LLaMA, GraphPrompter, GraphGPT, LLaGA) in terms of model size, training time, and inference time. While its computational cost exceeds traditional GNN methods, this is offset by its superior performance. Moreover, SimpleCIL also achieves competitive efficiency, presenting running-up performance with computational costs similar to the GNN-based method TPP. These results suggest that well-designed LLM approaches can balance performance and computational cost effectively.

Table 11: Performance comparison of GNN-, LLM-, and GLM-based methods in the DIL setting. Domain 1, Domain 2, and Domain 3 refer to the model continually learn on the PubMed, CiteSeer, and Cora datasets, respectively. The best and second results are highlighted.

| Methods | Domain 1 | | Domain 2 | | Domain 3 | | $\bar{\mathcal{A}}$ |
|---|---|---|---|---|---|---|---|
| | $\mathcal{A}$ | $\mathcal{F}$ | $\mathcal{A}$ | $\mathcal{F}$ | $\mathcal{A}$ | $\mathcal{F}$ | |
| GCN | $86.75_{\downarrow 0.00}$ | - | $28.26_{\downarrow 0.00}$ | $-58.49_{\downarrow 0.00}$ | $20.46_{\downarrow 0.00}$ | $-66.29_{\downarrow 0.00}$ | $45.16_{\downarrow 0.00}$ |
| EWC | $86.00_{\downarrow 0.75}$ | - | $27.38_{\downarrow 0.88}$ | $-59.37_{\downarrow 0.88}$ | $20.16_{\downarrow 0.30}$ | $-66.59_{\downarrow 0.30}$ | $44.52_{\downarrow 0.64}$ |
| Cosine | $85.50_{\downarrow 1.25}$ | - | $34.91_{\uparrow 6.65}$ | $-51.84_{\uparrow 6.65}$ | $27.41_{\uparrow 6.95}$ | $-59.34_{\uparrow 6.95}$ | $49.28_{\uparrow 4.12}$ |
| BERT | $89.67_{\uparrow 2.92}$ | - | $29.63_{\uparrow 1.37}$ | $-60.04_{\downarrow 1.55}$ | $18.17_{\downarrow 2.29}$ | $-71.58_{\downarrow 5.29}$ | $45.82_{\uparrow 0.66}$ |
| RoBERTa | $93.33_{\uparrow 6.58}$ | - | $30.27_{\uparrow 2.01}$ | $-63.06_{\downarrow 4.57}$ | $17.31_{\downarrow 3.15}$ | $-76.02_{\downarrow 9.73}$ | $46.97_{\uparrow 1.81}$ |
| LLaMA | $89.50_{\uparrow 2.75}$ | - | $64.27_{\uparrow 36.01}$ | $-22.48_{\uparrow 36.01}$ | $27.74_{\uparrow 7.28}$ | $-61.72_{\downarrow 4.57}$ | $60.50_{\uparrow 15.34}$ |
| SimpleCIL | $90.00_{\uparrow 3.25}$ | - | $66.06_{\uparrow 37.80}$ | $-20.69_{\uparrow 37.80}$ | $53.79_{\uparrow 33.33}$ | $-32.97_{\uparrow 33.33}$ | $69.95_{\uparrow 24.79}$ |
| GCN$_{\text{LLMEmb}}$ | $81.83_{\downarrow 4.92}$ | - | $39.31_{\uparrow 11.05}$ | $-42.44_{\uparrow 11.05}$ | $25.58_{\uparrow 5.12}$ | $-61.17_{\uparrow 5.12}$ | $48.91_{\uparrow 3.75}$ |
| ENGINE | $91.10_{\uparrow 4.35}$ | - | $34.17_{\uparrow 5.91}$ | $-56.93_{\downarrow 1.56}$ | $21.73_{\uparrow 1.27}$ | $-69.37_{\downarrow 3.08}$ | $49.00_{\uparrow 3.84}$ |
| GraphPrompter | $83.67_{\downarrow 3.08}$ | - | $52.22_{\uparrow 23.96}$ | $-31.45_{\uparrow 23.96}$ | $32.25_{\uparrow 11.79}$ | $-51.42_{\uparrow 14.87}$ | $56.08_{\uparrow 10.92}$ |
| GraphGPT | $91.17_{\uparrow 4.42}$ | - | $51.76_{\uparrow 23.50}$ | $-34.99_{\uparrow 23.50}$ | $28.86_{\uparrow 8.40}$ | $-62.31_{\uparrow 3.98}$ | $57.26_{\uparrow 12.10}$ |
| LLaGA | $85.50_{\downarrow 1.25}$ | - | $49.04_{\uparrow 20.78}$ | $-36.71_{\uparrow 20.78}$ | $30.39_{\uparrow 9.93}$ | $-55.36_{\uparrow 10.93}$ | $54.97_{\uparrow 9.81}$ |
| SimGCL(Ours) | $86.67_{\downarrow 0.08}$ | - | $\mathbf{74.53}_{\uparrow 46.27}$ | $\mathbf{-12.21}_{\uparrow 46.27}$ | $\mathbf{71.40}_{\uparrow 50.94}$ | $\mathbf{-15.06}_{\uparrow 50.94}$ | $\mathbf{77.54}_{\uparrow 32.38}$ |

Table 12: Time efficiency comparison of GNN-, LLM-, and GLM-based methods on Cora dataset. The asterisk (*) refers to the parameters that are only trainable during the initial session and remain fixed throughout subsequent sessions.

| Method | Param. (MB) | Training (s) | Inference (s) | $\bar{\mathcal{A}}$ | $\mathcal{A}_N$ |
|---|---|---|---|---|---|
| GCN | 0.19 | 4.76 | 0.02 | 57.0 | 38.2 |
| EWC | 0.19 | 5.66 | 0.02 | 56.0 | 31.0 |
| LwF | 0.19 | 7.75 | 0.03 | 55.7 | 30.8 |
| Cosine | 0.19* | 1.94 | 0.04 | 65.4 | 45.2 |
| TPP | 0.20 | 57.70 | 0.07 | 45.7 | 13.7 |
| BERT | 0.29 | 56.67 | 1.30 | 56.0 | 29.9 |
| RoBERTa | 1.76 | 172.85 | 4.16 | 54.6 | 29.6 |
| LLaMA | 3.25 | 454.47 | 72.65 | 65.6 | 53.8 |
| SimpleCIL | 1.76* | 55.90 | 4.04 | 70.8 | 58.3 |
| GCN$_{\text{LLMEmb}}$ | 0.53 | 5.18 | 0.03 | 59.1 | 31.1 |
| ENGINE | 3.19 | 89.70 | 0.11 | 59.2 | 31.3 |
| GraphPrompter | 10.65 | 805.15 | 99.34 | 61.9 | 46.8 |
| GraphGPT | 8.85 | 1247.46 | 584.72 | 55.5 | 31.6 |
| LLaGA | 22.04 | 688.78 | 425.59 | 58.2 | 30.2 |
| SimGCL(Ours) | 3.27* | 420.00 | 52.67 | 84.6 | 80.0 |

## E.5 VISUALIZATION

To further investigate the effectiveness of SimGCL, we performed a visualization experiment comparing the embeddings and prototypes generated by three prototype-based methods: Cosine, SimpleCIL, and SimGCL. The results are presented in Figure 4.

The results indicate that embeddings produced by SimGCL exhibit larger inter-class distances and smaller intra-class distances. Samples belonging to the same class are closely clustered around their respective prototypes, while distinct classes are separated by clear decision boundaries. In contrast, the embeddings generated by Cosine and SimpleCIL are noticeably more dispersed. In some cases, prototypes of different classes appear in close proximity or even overlap (as shown in the middle-bottom and top-left subfigures), suggesting that these methods may struggle to differentiate

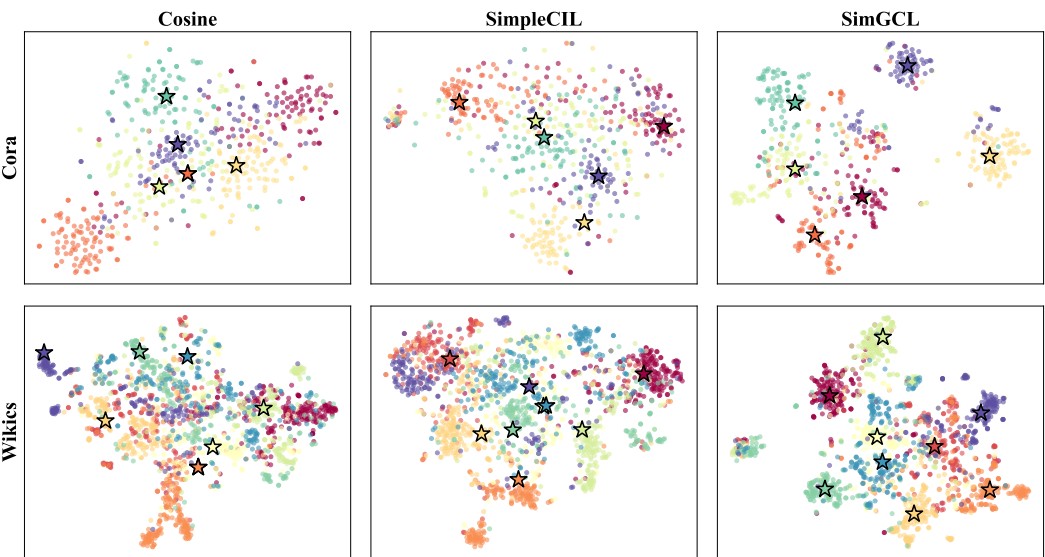

Figure 4: Visualization of the learnt embeddings and prototypes across three prototype-based methods.

samples from different classes. These findings underscore the superior embedding capability of SimGCL, reflecting its enhanced ability to comprehend both graph structures and textual information.

### E.6 PERFORMANCE EVOLUTION

To have deeper insights into the comparative learning dynamics, we present performance evolution curves in Figure 5. The visualization demonstrates SimGCL's consistent superiority over baseline models across multiple training sessions, underscoring its robust capability to maintain previously acquired knowledge while effectively assimilating new information.

### E.7 EWC WITH VARIOUS BACKBONES

Table 13: Performance comparison of different backbones integrated with EWC on Cora, WikiCS, and Arxiv datasets. "w/o" and "w/" represent learning without and with EWC, respectively.

| Dataset | Metric | GCN | | BERT | | RoBERTa | | LLaMA | | SimGCL |
|---|---|---|---|---|---|---|---|---|---|---|
| | | w/o | w/ | w/o | w/ | w/o | w/ | w/o | w/ | |
| Cora | $\bar{A}$ | 57.0 | 56.0 $_{\downarrow 1.00}$ | 56.0 | 55.5 $_{\downarrow 0.50}$ | 54.6 | 55.1 $_{\uparrow 0.50}$ | 65.6 | 62.1 $_{\downarrow 3.50}$ | 84.6 |
| | $A_N$ | 38.2 | 31.0 $_{\downarrow 7.20}$ | 29.9 | 30.0 $_{\uparrow 0.10}$ | 29.6 | 30.0 $_{\uparrow 0.40}$ | 53.8 | 49.4 $_{\downarrow 4.40}$ | 80.0 |
| WikiCS | $\bar{A}$ | 52.4 | 45.9 $_{\downarrow 6.50}$ | 58.4 | 54.9 $_{\downarrow 3.50}$ | 55.1 | 54.0 $_{\downarrow 1.10}$ | 55.5 | 54.5 $_{\downarrow 1.00}$ | 73.5 |
| | $A_N$ | 30.2 | 28.5 $_{\downarrow 1.70}$ | 30.0 | 30.4 $_{\uparrow 0.40}$ | 30.8 | 30.6 $_{\downarrow 0.20}$ | 30.9 | 30.8 $_{\downarrow 0.10}$ | 61.9 |
| Arxiv | $\bar{A}$ | 54.9 | 55.3 $_{\uparrow 0.40}$ | 26.0 | 24.1 $_{\downarrow 1.90}$ | 24.4 | 24.3 $_{\downarrow 0.10}$ | 24.9 | 23.8 $_{\downarrow 1.10}$ | 59.9 |
| | $A_N$ | 34.9 | 33.0 $_{\downarrow 1.90}$ | 9.7 | 1.3 $_{\downarrow 8.40}$ | 1.4 | 1.4 $_{\uparrow 0.00}$ | 1.4 | 1.2 $_{\downarrow 0.20}$ | 33.8 |

To further investigate the rationale and efficacy of integrating LLMs with traditional continual learning methods, we conducted experiments combining GCN and three LLM backbones—BERT, RoBERTa, and LLaMA—with EWC. Detailed results are provided in Table 13.

The results indicate that only the RoBERTa+EWC configuration on the Cora dataset yields a marginal performance improvement, suggesting the limited effectiveness of EWC when applied to LLMs. We attribute this to the extensive parameter space characteristic of LLMs: EWC relies on regularizing important parameters via the Fisher information matrix, which becomes computationally intractable at such a scale. Consequently, it is challenging to effectively constrain all critical parameters during continual learning, thereby reducing the method's practical utility.

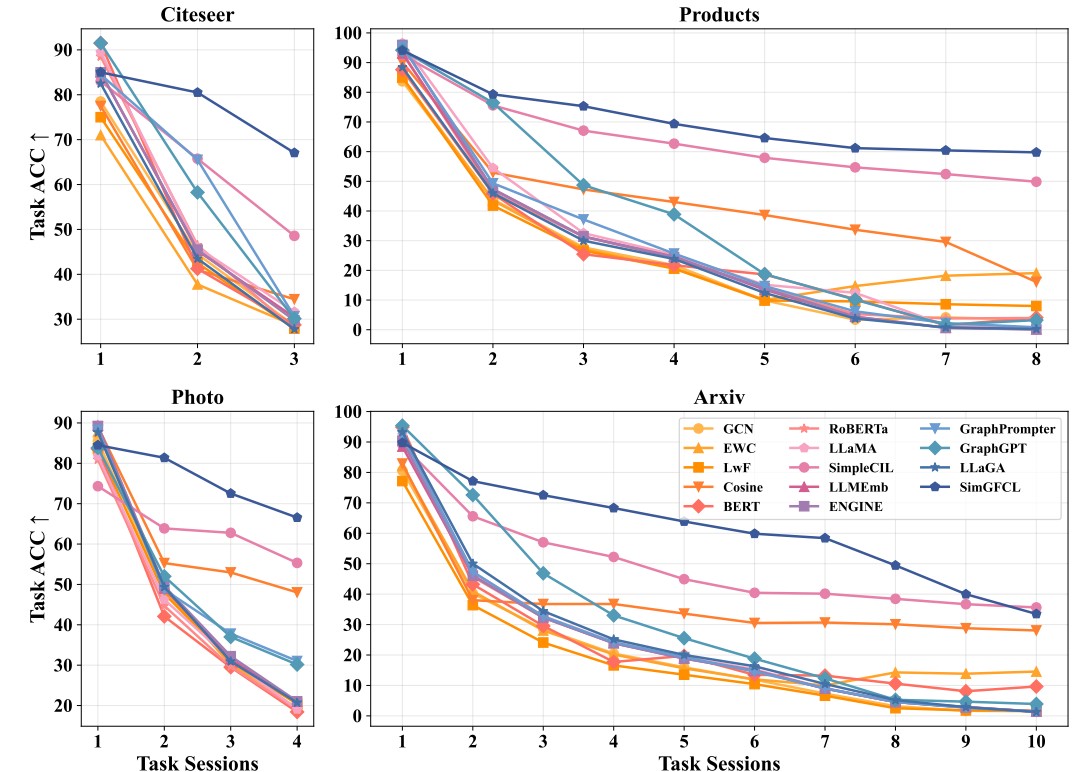

Figure 5: Performance evolution of all models on Citeseer, Photo, Products, and Arxiv datasets.

# F BROADER DISCUSSION

## F.1 LIMITATIONS

**Restricted to node-level tasks.** Currently, `LLM4GCL` is limited to GCL for node classification tasks, as most GLM-based methods are designed for single-task scenarios, and node classification remains the predominant task in GML. However, GML encompasses other critical tasks such as link prediction and graph classification. Moreover, in practical applications, GCL has significant value in these task domains, like predicting interactions between existing and new users (edge-level class incremental learning, ECIL) and identifying novel protein categories (graph-level class incremental learning, GCIL). Notably, recent advances in Graph Foundation Models (Mao et al., 2024) have enabled Graph-enhanced LLMs like OFA (Liu et al., 2023a), UniGraph (He et al., 2024), and UniAug (Tang et al., 2024b) to support edge-level or graph-level tasks. Therefore, a comprehensive evaluation of GLM-based methods in ECIL and GCIL scenarios is imperative, and we designate this as one of the key directions for our future research.

**Restricted to TAGs.** In `LLM4GCL`, all seven benchmark datasets are TAGs, as the evaluated GLM-based methods inherently depend on textual node attributes. However, this requirement limits applicability to real-world non-textual graphs (e.g., transportation networks or flight routes). Currently, emerging solutions like GCOPE (Zhao et al., 2024a) and SAMGPT (Yu et al., 2025) demonstrate promising approaches for text-free graph foundation models. Consequently, extending GLM-based methods to such non-textual graph scenarios remains a critical direction for future research.

## F.2 FURTHER DISCUSSION

**Apply SimGCL to extreme scenarios.** We observe that SimGCL exhibits relatively inferior performance on extremely sparse graphs, under very few-shot conditions, and in long session scenarios. We attribute this to its reliance on informative ego-graph prompts and extensive pre-training during the initial session. Sparse graphs provide limited structural context, which may result in less informa-

tive prompts and hinder the model's graph comprehension. Furthermore, in FSNCIL settings, the substantial data disparity between the base session and incremental sessions introduces a bias toward base classes, increasing the risk of overfitting.

Nonetheless, SimGCL represents a flexible framework that can be adapted to these challenging settings through targeted improvements: for sparse graphs, integrating lightweight graph encoders such as SGC (Wu et al., 2019) could enhance higher-order structural awareness; for few-shot learning, session-aware fine-tuning via LoRA (Hu et al., 2022) with momentum updates (He et al., 2020) could mitigate overfitting to the base session; for long sessions, prototype calibration methods (Li et al., 2025a) may help maintain stability across old and new classes.

In summary, although SimGCL shows certain limitations in highly challenging environments, it successfully provides a versatile foundation for graph continual learning, as its performance shortcomings can be alleviated through context-aware enhancements.

**Generalization of GLM-based Methods.** Current GLM-based methods generally demonstrate unsatisfactory performance compared to pure LLM-based methods, facing challenges in GCL tasks, yet their superior generalization and contextual understanding capabilities - surpassing shallow GNNs - show significant potential for complex GCL scenarios when properly implemented. For example, SimpleCIL outperforms most GNN and LLM baselines by leveraging class prototypes, fully utilizing LLMs' generalization while reducing overfitting; similarly, SimGCL significantly outperforms existing models by being trained with informative graph structural prompts, establishing new SOTA performance across multiple datasets. Consequently, we believe that this paper demonstrates a promising direction for leveraging LLMs/GLMs in GCL, laying the groundwork for unlocking their full potential in this emerging research direction.

### F.3 POTENTIAL IMPACTS

The integration of LLMs with graph tasks represents an emerging and highly promising research direction, demonstrating substantial potential across diverse applications and particularly in GCL scenarios. This paper proposes `LLM4GCL` to advance research focus on this novel paradigm that leverages graph-enhanced LLMs for improved GCL performance. Through `LLM4GCL`, we conduct a systematic investigation into graph-enhanced LLM methodologies and establish comprehensive benchmarking across various graph domains. We believe this work will serve as the catalyst and pioneer for accelerated progress in this developing research community.

Moreover, as `LLM4GCL` demonstrates several efficient LLM- and GLM-based GCL methods, we argue that these advancements hold significant societal value. Specifically, the proposed methods facilitate efficient knowledge updates in dynamic environments, such as social networks and recommendation systems, through seamless integration with LLMs, thereby substantially reducing computational overhead compared to conventional training-from-scratch paradigms. Furthermore, the cross-domain adaptability of these models offers distinct advantages in mission-critical applications, including healthcare diagnostics and financial forecasting. In such domains, continuous model adaptation to emergent knowledge, ranging from novel therapeutic interventions to evolving market trends, can be achieved while effectively mitigating catastrophic forgetting. However, practical deployment necessitates careful consideration of several key challenges, including data privacy implications, the potential for bias propagation in automated decision-making processes, and ensuring robustness against adversarial data injection, particularly in open-domain applications.

In the future, we will keep track of newly emerged techniques in the GCL field and continuously update `LLM4GCL` with more solid experimental results and detailed analyses.

