# OpenReview forum: "Can LLMs Alleviate Catastrophic Forgetting in Graph Continual Learning? A Systematic Study"
_ICLR.cc/2026/Conference — Submitted to ICLR 2026_

### Official Review · Reviewer_FyMh · 2025-10-18

**Soundness:** 3
**Presentation:** 3
**Contribution:** 2
**Rating:** 4
**Confidence:** 4

**Summary:**

This paper investigates whether large language models (LLMs) can help overcome catastrophic forgetting in Graph Continual Learning (GCL) tasks.

- They have identified a serious flaw, task ID leakage, in common GCL evaluations, where models inadvertently access task identifiers during testing.

- They introduce LLM4GCL, the first systematic benchmark for LLMs in GCL.

- Proposed a method, SimGCL, that consistently achieves state-of-the-art results and mitigates forgetting effectively.

**Strengths:**

### 1. Theoretical Soundness
- The work is well motivated as the authors have identified that all prior GCL methods train from scratch and do not exploit pretrained models’ generalization.

- Uses Parameter-Efficient Fine-Tuning (PEFT) via LoRA, a proven method for adapting large models efficiently.

- Employs prototype-based classification, which is widely accepted as a way to mitigate forgetting in continual learning.

- The graph prompt design is conceptually consistent with instruction tuning paradigms in multimodal LLMs.

### 2. Experimental Soundness
- The paper introduces LLM4GCL, with 9 LLM/GLM methods and 7 datasets. This is a substantial empirical foundation for a GCL.

- The identification of task ID leakage in existing setups is a major contribution.

- They release code, datasets, and standardized evaluation. This is a strong signal of methodological transparency.

**Weaknesses:**

### 1. Theoretical Limitations
- They assume graph structure can be effectively encoded as text prompts. While innovative, this verbalization step might oversimplify structural relationships, potentially losing fine-grained topology information.

- Use of LoRA only in the first session assumes that continual adaptation can be achieved purely through frozen embeddings and prototype updates — a strong assumption that may not generalize across highly dynamic graph evolutions.

- Missing formal theoretical analysis of why the prototype mechanism preserves generalization in LLM-embedded graph representations (e.g., no bounds on forgetting or feature drift).

### 1. Experimental Limitations
- The authors do not deeply dissect which component of SimGCL contributes most (LoRA, prototype, or prompt design). Without that, causal claims (“graph prompts mitigate forgetting”) remain partially speculative.

- In small-scale datasets like Cora and Citeseer, very large LLMs might overfit to textual node attributes, inflating reported improvements.

**Questions:**

Please refer to the weakness section. I would urge the authors to pay special attention to the following concerns first.

- The work is conceptually sound and empirically motivated but not theoretically grounded. The rationale aligns with existing literature, but lacks formal guarantees. Can the authors provide some theoretical justifications to their method?

- Empirical claims are mostly well-supported, though some causal explanations (e.g., why GLMs fail) remain conjectural. Therefore, I request the authors to substantiate such claims.

- Address limited ablation analyses and potential structural oversimplifications when converting graph topology into text prompts.

---

> ### Author Response · Authors · 2025-11-25
> **Response [Part 1]**
>
> > W1. They assume graph structure can be effectively encoded as text prompts. While innovative, this verbalization step might oversimplify structural relationships, potentially losing fine-grained topology information.
>
> Thank you for your insightful comments. We agree that there might exist risk of oversimplification, but we argue that **the utilization of graph prompt is a worthwhile trade-off**.
>
> While mainstream GLM methods often achieve strong structural representation abilities by integrating with GNNs or learnable structural vectors, which is a key factor leading to their SOTA performance on **static** node classification, they face severe performance degradation in **dynamic** GCL setting. This is because the shallow GNN model or learnable structural vectors undergo drastic parameter changes across multiple GCL sessions, exacerbating the misalignment between the graph and LLM text representations. In contrast, by employing neighborhood-aware prompts that include information up to 2-hop neighbors, the potential loss of structural detail in SimGCL can be effectively mitigated without any parameter updation in incremental sessions, which better perserves LLMs' knowledge and abilities. The empirical results, showing an absolute increase of nearly 20% over the previous GNN state-of-the-art baselines, confirm that this graph structure based prompt approach is a more effective pathway to integrate LLMs with graph knowledge.
>
> For future work, we will explore more efficient integration methods to enhance LLMs' higher-order structural awareness, mitigating the structural information loss.
>
> > W2. Use of LoRA only in the first session assumes that continual adaptation can be achieved purely through frozen embeddings and prototype updates — a strong assumption that may not generalize across highly dynamic graph evolutions.
>
> Thank you for your comments. We would like to clarify that the LoRA tuning in the first session is implemented to efficiently adapt the LLM's knowledge to the graph task's feature space, without destroying the LLM's generalization abilities. Full parameter fine-tuning, conversely, will pose a much greater risk of damaging the model's parameters and compromising generalization in subsequent sessions. Furthermore, the prototype network is a widely accepted, training-free approach used to mitigate catastrophic forgetting and maintain model generalization capacity [1,2].
>
> In our view, the scenario you mentioned regarding "highly dynamic graph evolutions", is closely related to the DIL setting, where graph structure and feature distribution significant differences across sessions. Based on this scenario, our extended experiments (Table 11, Appendix E.3) demonstrate that SimGCL achieves state-of-the-art results in the DIL setting, confirming the ability of our frozen-backbone approach to generalize robustly even in highly dynamic scenarios.
>
> In future work, we will explore applying task- or session-specific adjustments to the prototypes, such such as prototype calibration methods, to increase the model's adaptivity to the most extreme dynamic evolutions.
>
> [1] Zhou, Da-Wei, et al. "Revisiting class-incremental learning with pre-trained models: Generalizability and adaptivity are all you need." International Journal of Computer Vision 133.3 (2025): 1012-1032.
>
> [2] Li, Yayong, et al. "Inductive graph few-shot class incremental learning." Proceedings of the Eighteenth ACM International Conference on Web Search and Data Mining. 2025.
>
> > W3. Missing formal theoretical analysis of why the prototype mechanism preserves generalization in LLM-embedded graph representations (e.g., no bounds on forgetting or feature drift).
>
> Thank you for your careful reviewing. Here we would like to clarify that the primary goal of this paper is to establish the first "LLM for GCL" benchmark and propose an empirically effective method SimGCL for this scenario. Despite lacking formal theoretical bounds, we provide strong empirical evidence in our visualization results (Figure 4, Appendix E.5) to show that the prototype mechanism successfully preserves class-discriminative features by generating embeddings with larger inter-class distances and tighter intra-class clustering. SimGCL’s consistent SOTA performance across all metrics further validates its superior knowledge retention.
>
> In the future, we will focus on making formal theoretical analysis of prototype networks, including mathematically deriving bounds on feature drift and forgetting, for a better understanding of SimGCL's performance.

---

> ### Author Response · Authors · 2025-11-25
> **Response [Part 2]**
>
> > W4. The authors do not deeply dissect which component of SimGCL contributes most (LoRA, prototype, or prompt design). Without that, causal claims (“graph prompts mitigate forgetting”) remain partially speculative.
>
> |Model|Cora $\overline{\mathcal{A}}$|Cora $\mathcal{A}_N$|Citeseer $\overline{\mathcal{A}}$|Citeseer $\mathcal{A}_N$|WikiCS $\overline{\mathcal{A}}$|WikiCS $\mathcal{A}_N$|
> |:-|:-:|:-:|:-:|:-:|:-:|:-:|
> |Fine-tune|48.7|33.2|50.1|37.0|38.6|15.4|
> |LoRA|65.6|53.8|55.7|31.7|55.5|30.9|
> |LoRA+Proto.|78.5|65.4|70.6|57.6|60.3|54.4|
> |LoRA+Prompt|68.5|55.1|60.3|39.6|63.9|41.7|
> |LoRA+Prompt+Proto.|84.6|80.0|77.1|66.3|73.5|61.9|
>
> Thank you for your suggestions. Following your advice, we conduct an ablation study comparing the contribution of LoRA, graph prompt and prototype, the results are in the above table. It can be observed that, firstly, the performance of the model without LoRA tuning is significantly worse, as the full-parameter tuning will change more LLMs' paprameters during various sessions, leading to more severe catastophic forgetting and model performance degradation. Secondly, the contribution of the prototype mechanism to model performance is substantial. Without the prototype mechanism, the performance gain from graph prompt is limited, because the model still undergoes continuous parameter updates across sessions. However, the improvement becomes significant when the prompt is introduced alongside the prototype. We hypothesize this synergy exists because, although the graph prompt can provide richer information, it intensify the model's tendency toward overfitting. In contrast, by freezing the parameters, prototype network ensures model's stability, which allows the graph prompt to supply the auxiliary information needed for the model to make more accurate judgments in incremental sessions without parameter updation.
>
>
> > W5. In small-scale datasets like Cora and Citeseer, very large LLMs might overfit to textual node attributes, inflating reported improvements.
>
> Thank you for your comments. Here we would like to point out that the tendency to overfit is a universal challenge across all architectures in GCL. Our SimGCL method strategically counters this by using a training-free prototype classifier and parameter freezing, which prevents the catastrophic forgetting and overfitting resulting from constant parameter updates in incremental sessions.
>
> Moreover, SimGCL's superior performance is not limited to small datasets; it achieves SOTA results on large-scale and dense datasets like Products. This consistent superiority across diverse scales and densities proves that our mechanism facilitates robust continual learning, rather than merely exploiting local memorization.
>
> Furthermore, our model scaling study (Figure 3) demonstrates that increasing LLM parameters consistently enhances generalization and boosts GCL performance. If the large models were simply overfitting, this sustained performance gain should diminish or saturate, particularly in later sessions.

---

> ### Author Response · Authors · 2025-11-27
> **We are looking forward to your feedback**
>
> Dear Reviewer FyMh,
>
> Thank you again for your efforts in reviewing our manuscript and for providing constructive and insightful comments. Given that the rebuttal period remains only less than one week, we would be very grateful if you could briefly confirm whether our clarifications have resolved your main points. Moreover, if you have any remaining ambiguities or further questions persist, we are pleased to provide detailed elaboration.
>
> Thank you once more for your valuable engagement with our work.
>
> Sincerely,
>
> Authors of Submission 1449

---

### Official Review · Reviewer_FJeP · 2025-10-21

**Soundness:** 3
**Presentation:** 3
**Contribution:** 2
**Rating:** 4
**Confidence:** 5

**Summary:**

This paper focuses on the problem of graph continual learning and proposes a method based on LLMs. Different from the previous settings, the authors provide a new setting called global testing and construct a benchmark. Based on the benchmark, a new method based on LLMs and prototype learning is proposed.

**Strengths:**

1. The paper is well-written and easy to follow. The code is released.

2. A new benchmark is proposed to evaluate the performance of LLMs and graph-enhanced LLMs.

3. The authors identify the flaw of existing GCL settings and propose global testing. To utilize the generalization capacity of LLMs, a GLM-based approach is proposed.

**Weaknesses:**

1. The constraint of setting each task to have a unified sample size is unreasonable, and the label imbalance problem is the real challenge that should be addressed.

2. The introduction of global testing is not very clear. More clear figures or visualizations are needed.

3. The used graphs are all text-attribute graphs (TAGs). Can the proposed method be extended to non-TAGs?

4. Experimental details should be included in the main paper to provide readers with a general picture of the experimental settings.

5. There are some typos, and the authors should carefully revise the paper.

**Questions:**

Please see the weaknesses.

---

> ### Author Response · Authors · 2025-11-25
> **Response [Part 1]**
>
> > W1. The constraint of setting each task to have a unified sample size is unreasonable, and the label imbalance problem is the real challenge that should be addressed.
>
> Thank you for your insightful comments. First we would like to clarify that the motivation behind this setting is to establish a clear benchmark for continual learning and the catastrophic forgetting issue. Label imbalance is a very general problem exists in all classification tasks, consequently, in continual learning, if we do not avoid the label imbalance issue, the label imbalance problem will intertwine with GCL, making it difficult to disentangle the root causes of performance degradation. By removing classes with insufficient samples and utilizing a unified sample size, we isolate and focus strictly on the core continual learning challenge.
>
> Furthermore, in our experiments, this adjustment was minimal: we only removed classes with extremely few samples (e.g., classes with less than 10 samples) in the Products and Arxiv-23 datasets. Such extreme sparsity is inherently unsuitable for stable metric calculation and would negatively skew results analysis.
>
> We acknowledge that analyzing label imbalance is a crucial real-world problem. In future work, we will commit to specifically analyzing the impact of class imbalance on continual learning performance.
>
> > W2. The introduction of global testing is not very clear. More clear figures or visualizations are needed.
>
> Thank you for your comments. First, we would like to re-claim the global testing setup for you:
>
> ---
> **Scope of the Testing Graph**: When evaluating the $i$-th task, the test graph $\mathcal{G} _{q _{j}}$ utilized is the union of all subgraphs from the first task up to the current task $(\mathcal{G} _{q _{j}}=\bigcup _{z=1}^{j}\mathcal{G} _{s _{z}})$. This necessitates that the model, when predicting nodes in the current task, must consider nodes and edges related to all preceding tasks.
>
> **Scope of the Class Set**: The class set for testing, $\mathcal{C} _{q _{j}}$, is the union of all classes learned so far $(\mathcal{C} _{q _{j}}=\bigcup _{z=1}^{j}\mathcal{C} _{s _{z}})$. This forces the model to perform predictions across the complete, ever-growing class space, thereby genuinely testing its continual learning and anti-forgetting capabilities.
>
> ---
>
> Following your advice, we have made a comprehensive comparsion between the settings of local testing and global testing as follows:
>
> | |Local Testing|Global Testing|
> |:-:|:-:|:-:|
> |**Session 1 $\mathcal{T} _{s _1}$**|**Evaulates on $\|\mathcal{T} _{q _1}\|=1$**|**Evaulates on $\|\mathcal{T} _{q _1}\|=1$**|
> |Testing Graph $\mathcal{G} _{q _j}$|$\mathcal{G} _{q _1}=\mathcal{G} _{s _1}$|$\mathcal{G} _{q _1}=\mathcal{G} _{s _1}$|
> |Testing Classes $\mathcal{C} _{q _j}$|$\mathcal{C} _{q _1}=\mathcal{C} _{s _1}$|$\mathcal{C} _{q _1}=\mathcal{C} _{s _1}$|
> |**Session 2 $\mathcal{T} _{s _2}$**|**Evaulates on $\|\mathcal{T} _{q _2}\|=2$**|**Evaulates on $\|\mathcal{T} _{q _2}\|=1$**|
> |Testing Graph $\mathcal{G} _{q _j}$|$\mathcal{T} _{q _{2,1}}$: $\mathcal{G} _{q _{2,1}}=\mathcal{G} _{s _1}$; $\mathcal{T} _{q _{2,2}}$: $\mathcal{G} _{q _{2,2}}=\mathcal{G} _{s _2}$|$\mathcal{T} _{q _2}$: $\mathcal{G} _{q _2}=\mathcal{G} _{s _1}\cup\mathcal{G} _{s _2}$|
> |Testing Classes $\mathcal{C} _{q _j}$|$\mathcal{T} _{q _{2,1}}$: $\mathcal{C} _{q _{2,1}}=\mathcal{C} _{s _1}$; $\mathcal{T} _{q _{2,2}}$: $\mathcal{C} _{q _{2,2}}=\mathcal{C} _{s _2}$|$\mathcal{T} _{q _2}$: $\mathcal{C} _{q _2}=\mathcal{C} _{s _1}\cup\mathcal{C} _{s _2}$|
> |**Session 3 $\mathcal{T} _{s _3}$**|**Evaulates on $\|\mathcal{T} _{q _3}\|=3$**|**Evaulates on $\|\mathcal{T} _{q _3}\|=1$**|
> |Testing Graph $\mathcal{G} _{q _j}$|$\mathcal{T} _{q _{3,1}}$: $\mathcal{G} _{q _{3,1}}=\mathcal{G} _{s _1}$; $\mathcal{T} _{q _{3,2}}$: $\mathcal{G} _{q _{3,2}}=\mathcal{G} _{s _2}$; $\mathcal{T} _{q _{3,3}}$: $\mathcal{G} _{q _{3,3}}=\mathcal{G} _{s _3}$|$\mathcal{T} _{q _3}$: $\mathcal{G} _{q _3}=\mathcal{G} _{s _1}\cup\mathcal{G} _{s _2}\cup\mathcal{G} _{s _2}$|
> |Testing Classes $\mathcal{C} _{q _j}$|$\mathcal{T} _{q _{3,1}}$: $\mathcal{C} _{q _{3,1}}=\mathcal{C} _{s _1}$; $\mathcal{T} _{q _{3,2}}$: $\mathcal{C} _{q _{3,2}}=\mathcal{C} _{s _2}$; $\mathcal{T} _{q _{3,3}}$: $\mathcal{C} _{q _{3,3}}=\mathcal{C} _{s _3}$|$\mathcal{T} _{q _3}$: $\mathcal{C} _{q _3}=\mathcal{C} _{s _1}\cup\mathcal{C} _{s _2}\cup\mathcal{C} _{s _3}$|
>
> We will add this table into the next version of our paper.

---

> ### Author Response · Authors · 2025-11-25
> **Response [Part 2]**
>
> > W3. The used graphs are all text-attribute graphs (TAGs). Can the proposed method be extended to non-TAGs?
>
> We thank you for your deep insights. We would like to clarify that one of the key successes of SimGCL lies in leveraging the LLM's generalization abilities in conjunction with the prototype network to alleviate catastrophic forgetting. Consequently, we can extend the method to non-TAGs with minimal modification by replacing the text-injected graph prompt with structure embeddings.
>
> Specifically, we propose a two-step approach: First, a lightweight graph encoder (e.g., a shallow GCN[1] or SGC[2]) would be introduced as a feature extractor to convert non-textual features into structure-aware vectors. Second, this structure vector would be channeled through a projection head to generate aligned structural pseudo-text or structure embeddings, which would replace the current ego-graph-derived text-based prompts. This preserves the core architecture, allowing the LLM to utilize its powerful semantic processing capacity and the training-free prototype classifier to execute tasks.
>
> However, given that LLMs inherently process textual information, TAGs remain the foundational domain for most current GLM research. We note that most mainstream graph datasets currently include text attributes, and existing GLM benchmarks primarily evaluate on TAGs[3-6], this is why our benchmark focuses on this setting. We fully admit that the challenge of how to more effectively extend GLMs to non-TAGs remains a critical and open issue for mainstream GLM methods[7]. In the future, we will follow your advice to further explore benchmarking GLMs for GCL tasks on non-textual graphs.
>
> [1] Kipf, T. N. "Semi-supervised classification with graph convolutional networks." arXiv preprint arXiv:1609.02907 (2016).
>
> [2] Wu, Felix, et al. "Simplifying graph convolutional networks." International conference on machine learning. Pmlr, 2019.
>
> [3] Li, Yuhan, et al. "Glbench: A comprehensive benchmark for graph with large language models." Advances in Neural Information Processing Systems 37 (2024): 42349-42368.
>
> [4] Li, Zhuofeng, et al. "Teg-db: A comprehensive dataset and benchmark of textual-edge graphs." Advances in Neural Information Processing Systems 37 (2024): 60980-60998.
>
> [5] Wang, Yuxiang, et al. "Exploring graph tasks with pure llms: A comprehensive benchmark and investigation." arXiv preprint arXiv:2502.18771 (2025).
>
> [6] Wu, Xixi, et al. "When Do LLMs Help With Node Classification? A Comprehensive Analysis." arXiv preprint arXiv:2502.00829 (2025).
>
> [7] Yu, Xingtong, et al. "Samgpt: Text-free graph foundation model for multi-domain pre-training and cross-domain adaptation." Proceedings of the ACM on Web Conference 2025. 2025.
>
> > W4. Experimental details should be included in the main paper to provide readers with a general picture of the experimental settings.
>
> Thank you for your suggestion. Currently, these details can only be placed in the Appendix due to page limit constraints. We will integrate key experimental information, such as the dataset statistics (from Appendix B.3, Table 6) and the NCIL/FSNCIL setting comparisons (from Appendix B.1, Table 5), into the main text to provide readers with a general picture of our setup in the next version of our paper.
>
> > W5. There are some typos, and the authors should carefully revise the paper.
>
> Thank you for your careful review. We sincerely apologize for any typos or grammatical inconsistencies that may have affected the readability of our manuscript. We have thoroughly re-read and meticulously edited the entire paper, specifically addressing issues such as incorrect citations (e.g., for GraphPrompter), misspellings in abbreviations (e.g., GCIL, FSNCIL), and general word or grammatical errors.

---

> ### Author Response · Authors · 2025-11-27
> **We are looking forward to your feedback**
>
> Dear Reviewer FJeP,
>
> Thank you again for your efforts in reviewing our manuscript and for providing constructive and insightful comments. Given that the rebuttal period remains only less than one week, we would be very grateful if you could briefly confirm whether our clarifications have resolved your main points. Moreover, if you have any remaining ambiguities or further questions persist, we are pleased to provide detailed elaboration.
>
> Thank you once more for your valuable engagement with our work.
>
> Sincerely,
>
> Authors of Submission 1449

---

### Official Review · Reviewer_PWgw · 2025-10-29

**Soundness:** 2
**Presentation:** 3
**Contribution:** 2
**Rating:** 4
**Confidence:** 4

**Summary:**

This paper investigates whether large language models can alleviate catastrophic forgetting in graph continual learning (GCL). The authors identify flaws in existing GCL evaluation protocols—particularly task ID leakage—and propose a revised global testing setup. They introduce LLM4GCL, a new benchmark encompassing multiple GNN-, LLM-, and GLM-based methods across seven text-attributed graph datasets, and propose SimGCL that combines graph-structured prompts with prototype-based prediction. Experimental results indicate that SimGCL performs competitively under various GCL settings, suggesting that pretrained LLMs have potential for continual graph learning.

**Strengths:**

1. The paper tackles an underexplored question—whether large language models (LLMs) can mitigate catastrophic forgetting in graph continual learning.
2. A benchmark, LLM4GCL, is developed.

**Weaknesses:**

1. My biggest concern lies in the motivation. The primary goal of continual learning is to enable efficient adaptation under limited resources, whereas LLMs are inherently computationally expensive. It is unclear whether the significant computational cost introduced by LLMs can truly justify the efficiency-driven learning protocol that continual learning aims to achieve.

2. Following this point, I believe the comparison should include GCL methods that allocate additional parameters (expansion-based) or memory (experience-replay-based) to narrow the substantial computational gap between LLMs and traditional GNNs. However, the authors compare their method only with regularization-based approaches, which are the most resource-limited class of GCL methods. Furthermore, the comparisons are arguably unfair since the methods rely on different backbone architectures for graph prediction.

3. The authors acknowledge task ID leakage and address it by introducing a global testing protocol, which is commendable. Nonetheless, all the datasets used (e.g., Cora, Citeseer, WikiCS) are widely used text-attributed benchmarks that may overlap with the pretraining corpora of LLMs. The paper does not verify whether such overlap exists, leaving potential knowledge leakage from pretraining unaddressed.

4. I also find the fairness of the LLM4GCL benchmark setup questionable. The LLM baselines are not fine-tuned on the same datasets and therefore lack domain knowledge of LLM4GCL, making the comparison asymmetric. The proposed method appears less focused on mitigating catastrophic forgetting and more on task-specific instruction tuning, which may even result in overfitting to the given dataset. This is evidenced by the large performance gap between the proposed method and other LLM-based baselines on LLM4GCL, in contrast to the much smaller gap observed on standard public datasets. Overall, the comparisons and evaluations in this paper feel ambiguous, making it difficult to discern the central message or takeaway the authors intend to convey.

**Questions:**

N/A

---

> ### Author Response · Authors · 2025-11-25
> **Response [Part 1]**
>
> > W1. My biggest concern lies in the motivation. The primary goal of continual learning is to enable efficient adaptation under limited resources, whereas LLMs are inherently computationally expensive. It is unclear whether the significant computational cost introduced by LLMs can truly justify the efficiency-driven learning protocol that continual learning aims to achieve.
>
> Thank you for your insightful comments. First, we would like to clarify that the "limited resources" constraint in the context of Continual Learning primarily refers to the **inaccessibility of data and historical records**, as new knowledge and data arrives in **a streaming manner**. The storage cost of maintaining an ever-growing stream of data and historical records across an infinite time horizon, poses a far greater challenge to data storage than the computation required for the model itself. Conversely, a robust continual learning model, even with large parameters, can remain static until new classes appear, requiring only a fixed, predictable amount of memory and computational resources.
>
> Second, by successful architectural design, our work empirically demonstrates that LLM-based methods can effectively balance high performance with acceptable computational cost, thus achieving the efficiency goals of CL. Specifically, SimGCL minimizes computational overhead by concentrating the costly LLM adaptation into a single, initial step. It employs LoRA, only during the first training session. Moreover, for all subsequent incremental sessions, SimGCL utilizes a training-free prototype classifier for prediction, entirely avoiding the need for expensive parameter updates caused by the arrival of new knowledge. This design significantly alleviates the computational load during the incremental learning phase.
>
> Moreover, our time efficiency analysis (Appendix Table 12) empirically validates this strategy. SimGCL achieves superior continual learning performance compared to GNN-based models while exhibiting training overhead that is significantly less than GLM-based methods (e.g., GraphGPT), thus proving its viability and efficiency.
>
>
> > W2. Following this point, I believe the comparison should include GCL methods that allocate additional parameters (expansion-based) or memory (experience-replay-based) to narrow the substantial computational gap between LLMs and traditional GNNs. However, the authors compare their method only with regularization-based approaches, which are the most resource-limited class of GCL methods. Furthermore, the comparisons are arguably unfair since the methods rely on different backbone architectures for graph prediction.
>
> Thank you for your comments. As for the reply-based methods, our primary focus is strictly on the rehearsal-free constraint, as real-world GCL often prohibits access to previous task data due to privacy and storage limitations. Furthermore, the cost of perpetually storing vast historical data streams can be greater than the computational overhead introduced by LLM parameters. Therefore, we deliberately excluded all replay-based baselines from our evaluation for fairness comparsion. As for expansion-based methods, they are less common in classic GCL literature compared to regularization and replay techniques, and rarely appear as standard baselines in previous work, leading to their exclusion from our main consideration.
>
> Regarding the backbone architectures, the main objective of this work is precisely to investigate whether the LLM/GLM architecture can fundamentally promote better performance in GCL than traditional GNNs, necessitating the architectural difference. However, we ensured fairness by integrating with same backbone within architectural categories: GNN-based methods primarily use GCN, and GLM/LLM baselines primarily use LLaMA. We further ensured fairness by employing grid-search to find the optimal performance for each model on the same benchmark.

---

> ### Author Response · Authors · 2025-11-25
> **Response [Part 2]**
>
> > W3. The authors acknowledge task ID leakage and address it by introducing a global testing protocol, which is commendable. Nonetheless, all the datasets used (e.g., Cora, Citeseer, WikiCS) are widely used text-attributed benchmarks that may overlap with the pretraining corpora of LLMs. The paper does not verify whether such overlap exists, leaving potential knowledge leakage from pretraining unaddressed.
>
> Thank you for your deep insights. We acknowledge the potential for knowledge overlap is a critical, universal challenge for LLM-based models in general, however we would like to argue that it is not a determining factor in our experimental results.
>
> Firstly, if such an overlap were significantly aiding model performance, pure LLM-based models, without incorporating any graph structure, should achieve high performance. However, our results contradict this: pure LLM-based baselines (BERT, RoBERTa, and LLaMA) only achieved marginal improvements compared with GNN-based moedels. Furthermore, this observation is even starker in the FSNCIL setting, where LLM-based models often perform worse than the Vanilla GCN baseline. This suggests that potential overlap did not provide an artificial advantage in solving the complex continual learning task.
>
> Secondly, we emphasize that continual learning is not a task solvable by pre-trained knowledge alone; it demands a precise balance between generalization and stability. Over-relying on either aspect leads to suboptimal results. LLM-based and GLM-based methods, while benefiting from their large parameters and pre-trained knowledge to enhance generalization and reasoning, still exhibit a strong tendency to overfit the new tasks, which is similar to GNNs. This is evidenced by their minimal advantage in tasks with longer sessions, such as Arxiv. Consequently, in this paper we focus on architectural solutions and find stability-plasticity balance, which is precisely why prototype-based models achieved success.
>
>
> > W4. I also find the fairness of the LLM4GCL benchmark setup questionable. The LLM baselines are not fine-tuned on the same datasets and therefore lack domain knowledge of LLM4GCL, making the comparison asymmetric. The proposed method appears less focused on mitigating catastrophic forgetting and more on task-specific instruction tuning, which may even result in overfitting to the given dataset. This is evidenced by the large performance gap between the proposed method and other LLM-based baselines on LLM4GCL, in contrast to the much smaller gap observed on standard public datasets. Overall, the comparisons and evaluations in this paper feel ambiguous, making it difficult to discern the central message or takeaway the authors intend to convey.
>
> Thank you for your comments. Firstly all LLM and GNN baselines adhere to a consistent training protocol: all models are sequentially trained and evaluated on every session of each dataset. There are no specially tuned models or non-fine-tuned LLM baselines included. Thus, the training process is completely fair across all methods.
>
> Secondly, as for your concern about the overfitting issue, we would like to clarify that the performance of SimGCL stem from the synergistic effect of graph-structured instruction tuning and the prototype network, which achieves the crucial balance between generalization and stability: the instruction tuning aligns the LLM's textual understanding capabilities specifically to the graph-structured prompt space, while the prototype network preserves the LLM's strong generalization and inference capabilities by freezing the LLM's parameters after the initial session, preventing catastrophic forgetting that arises from overfitting during training on incremental sessions. **If SimGCL had suffered from severe overfitting in the first session, it would be unable to generate sufficiently discriminative feature representations for the new classes and sessions, leading to poor overall performance.** Moreover, as for other GLM-based baselines (e.g., GraphPrompter, GraphGPT), they also integrate with graph structure(e.g., by integrating with GNNs) and achieve competitive SOTA results on node classification tasks, yet they fail to achieve competitive performance on GCL tasks. This underperformance is precisely what we attribute to their inability to prevent overfitting on new tasks. Consequently, the central message of our paper is: how to find the optimal balance between generalization and stability in LLMs. SimGCL achieves this by coupling graph knowledge with a prototype-based stability mechanism, thus mitigating the overfitting problem in GCL.

---

> ### Author Response · Authors · 2025-11-27
> **We are looking forward to your feedback**
>
> Dear Reviewer PWgw,
>
> Thank you again for your efforts in reviewing our manuscript and for providing constructive and insightful comments. Given that the rebuttal period remains only less than one week, we would be very grateful if you could briefly confirm whether our clarifications have resolved your main points. Moreover, if you have any remaining ambiguities or further questions persist, we are pleased to provide detailed elaboration.
>
> Thank you once more for your valuable engagement with our work.
>
> Sincerely,
>
> Authors of Submission 1449

---

### Official Review · Reviewer_Sw4z · 2025-10-29

**Soundness:** 2
**Presentation:** 2
**Contribution:** 2
**Rating:** 2
**Confidence:** 4

**Summary:**

The paper proposes an extension of the Online Graph Learning problem formulation, aiming to address some of the limitations typically found in existing literature. Based on this revised formulation, the authors define two scenarios: a standard setting and a few-shot setting. They evaluate both conventional graph-based approaches and LLM-based methods. Finally, the paper introduces SimCML, a model that leverages a single round of instruction tuning to directly produce class prototypes from the embeddings generated by its fine-tuned LLM.

**Strengths:**

The paper explores a novel approach that connects LLMs with the graph continual learning framework. Moreover, it analyzes the commonly used GCL scenario, revealing an interesting limitation and proposing a solution to address this issue.

**Weaknesses:**

The paper presents several issues, mainly related to the clarity of presentation and the design of the experimental framework.
Regarding the presentation, the paper does not clearly explain the proposed model or the models used for comparison. As a result, many methodological choices are not properly justified. Going into more detail, in the introduction, the authors claim that there is a lack of investigation into the rationality of the common experimental setup used for GCL. It is worth noting that a recent study has already explored this aspect [1]. Another meaningful reference that should be considered and discussed is [6].
In the background section, the authors correctly highlight one of the main limitations of the commonly adopted GCL setting, which I agree with, and propose the idea of global testing. However, it is not clear how this proposal relates to the subsequent discussion about inter-task edges, which the authors claim not to consider in their tasks. This choice isolates subgraphs, making the problem closer to the original formulation. This point should be discussed and clarified in greater depth. Furthermore, the removal of the limited-representation classes appears to be a strong design choice that requires a more thorough discussion, particularly regarding its advantages and potential drawbacks.
In Section 3.2, the authors claim that the considered datasets cover a wide range of possible sizes. However, in practice, the investigation is limited to relatively small datasets. Large-dimensional datasets, such as OGN-Products and Reddit [2], are not considered. Moreover, most of the selected datasets are citation networks. Considering that the goal of the paper is to establish a benchmark for testing models in GCL, the dataset selection appears rather limited. Another important distinction that can significantly impact the GCL scenario is the difference between heterophilous and homophilous datasets, which is not addressed in this work.
In the paper, both the definitions of the literature methods and that of the proposed SimCML are not presented clearly, which makes it difficult to follow the flow of the discussion and to evaluate the novelty of the proposed approach. For instance, the discussion of the LLM baseline is very limited, the authors only state that it is nascent and that no methods have yet been specifically designed for CGL tasks. What remains unclear is how the LLM baselines are actually applied in the context of CGL, and consequently how the reported results are obtained. Regarding the models considered, it is surprising that the authors do not include several of the most promising replay-based approaches, such as CaT [3], PDGNN [4], SSM [5], among others. Another major issue is the lack of explanation regarding the experimental setting and the validation policy. While the appendix lists the hyperparameters considered, many of them are fixed, and it is not clear how these values were selected. Considering the aim of the paper, this represents a significant limitation. Moreover, the results reported in the tables do not include any measures of variance or standard deviation, raising concerns about whether the experiments were run only once. If this is the case, it is unclear how the authors assess the stability of the results or the statistical significance of the differences between models. This is particularly important for the LLM application, where variance in results is typically more pronounced.


[1] Donghi, G., Pasa, L., Zambon, D., Alippi, C. and Navarin, N., 2025. Online Continual Graph Learning. arXiv preprint arXiv:2508.03283.

[2] Zhang, X., Song, D. and Tao, D., 2024. Continual learning on graphs: Challenges, solutions, and opportunities. arXiv preprint arXiv:2402.11565.

[3] Liu, Y., Qiu, R. and Huang, Z., 2023, December. Cat: Balanced continual graph learning with graph condensation. In 2023 IEEE International Conference on Data Mining (ICDM) (pp. 1157-1162). IEEE.

[4] Zang, X., Song, D., Chen, Y. and Tao, D., 2024, August. Topology-aware embedding memory for continual learning on expanding networks. In Proceedings of the 30th ACM SIGKDD Conference on Knowledge Discovery and Data Mining (pp. 4326-4337).

[5] Zhang, X., Song, D. and Tao, D., 2022, November. Sparsified subgraph memory for continual graph representation learning. In 2022 IEEE International Conference on Data Mining (ICDM) (pp. 1335-1340). IEEE.

[6] Huang, S., Parviz, A., Kondrup, E., Yang, Z., Ding, Z., Bronstein, M., Rabbany, R. and Rabusseau, G., 2025. Are Large Language Models Good Temporal Graph Learners?. arXiv preprint arXiv:2506.05393.

**Questions:**

-Could the authors clarify and provide more details on the proposed SimCML model and the LLM baselines, particularly how these baselines are applied in the context of GCL?

-Could the authors provide more information on how the hyperparameters were selected and the validation protocol used? Additionally, were the experiments repeated to assess the stability and statistical significance of the results?

-Considering the goal of establishing a benchmark for GCL, could the authors comment on the choice of datasets and whether including large-dimensional datasets or heterophilous/homophilous distinctions might affect the evaluation?

-Could the authors discuss the rationale for not including certain replay-based approaches, such as CaT, PDGNN, and SSM, and how their inclusion might impact the evaluation of SimCML?

---

> ### Author Response · Authors · 2025-11-25
> **Response [Part 1]**
>
> > W1. Regarding the presentation, the paper does not clearly explain the proposed model or the models used for comparison. As a result, many methodological choices are not properly justified.
>
> Thank you for your careful reviewing. We would like to clarify that we have indeed compiled a comprehensive summary and comparison of all baseline categories (GNN-based, LLM-based, and GLM-based methods), detailing their backbones and input formats within Table 7, Appendix C.2.
>
> We sincerely apologize for the ambiguity in the current manuscript. We will integrate this detailed overview of the comparison models and their specifications directly into the main body of the paper in the next version of our manuscript.
>
> > W2. Going into more detail, in the introduction, the authors claim that there is a lack of investigation into the rationality of the common experimental setup used for GCL. It is worth noting that a recent study has already explored this aspect [1]. Another meaningful reference that should be considered and discussed is [6].
>
> Thank you for your insightful suggestions. As for the paper "Online Continual Graph Learning"(OCGL), we acknowledge that they also regard the real-world graph data comes as the "streaming manner", and their study indeed offers new insights and more rigorous constraints for GCL paradigm. However, we assert that the focus and motivation of our setup analysis are distinct from the cited OCGL. Specifically, OCGL introduces the Online Continual Graph Learning setting and focuses on addressing the computational challenge of neighborhood expansion in GNNs during streaming processing, formalizing efficiency constraints. In contrast, our paper's analysis is focused on the validity of the existing evaluation metrics and protocols in session-based CIL. We fundamentally questioned whether reported model performance is truly reliable, finding that the prevalent Local Testing setup suffers from Task ID Leakage. Consequently, we believe there is no big contribution overlap between our paper and OCGL. Crucially, we also find that the evaluation of OCGL appears to **still rely on a form of local testing** and does not explicitly address the class-incremental task ID leakage problem we identified. This observation further confirms the necessity of our paper.
>
> However, considering this new paper, we do acknowledge their contribution to the GCL scenario, and we agree to revise our Introduction to more accurately reflect our paper's contribution on the GCL setup rationality.
>
> As for the paper "Are Large Language Models Good Temporal Graph Learners?"(TGTalker) you have mentioned, we find common ground in the observation that both studies leverage the inherent strong generalization and reasoning capabilities of LLMs in graph domains, and both approaches employ prompt-based strategies to encode complex graph information for LLM processing, improving model adaptation to their specific tasks. However, we must first emphasize that our respective research domains and core challenges are fundamentally distinct.
>
> Our paper focuses on GCL, specifically addressing the **catastrophic forgetting challenge**, and the primary concern for SimGCL is maintaining both adaptivity to new classes and preservation of knowledge from old tasks as the task sequence grows. In contrast, TGTalker operates in the Temporal Graph Learning (TGL) domain, whose core challenge lies in accurately modeling temporal dependency, and time-evolving structure to predict the next link interaction, it can be regarded as that TGTalker only needs to continously adapt to new patterns without the concern to keep the previous knowledge, a completely different objective from GCL.
>
> Methodologically, we also find some slight difference between SimGCL and TGTalker. SimGCL utilizes graph-based instruction tuning and meticulously incorporate rich ego-graph derived prompts encompassing up to 2-hop neighborhood structure to instill comprehensive topological knowledge into the LLM. Conversely, the TGTalker framework is designed to be training-free, relying entirely on In-Context Learning (ICL). While this is effective for temporal prediction, its reliance on ICL and less alignment with the graph structure scenario makes it a concern for TGTalker's graph learning abilities.

---

> ### Author Response · Authors · 2025-11-25
> **Response [Part2]**
>
> > W3. It is not clear how this proposal relates to the subsequent discussion about inter-task edges, which the authors claim not to consider in their tasks. This choice isolates subgraphs, making the problem closer to the original formulation. This point should be discussed and clarified in greater depth.
>
> Thank you for your careful reviewing. Our choice to exclude inter-task edges during training follows previous work[1] and is necessary to maintain the rehearsal-free constraint. Since GNN's message-passing inherently aggregates neighbor information, inter-task edges would potentially expose previous task knowledge, which is prohibited in our scenario. However, it should be noted that inter-task edges are only excluded during **training phase**. In the global testing phase, the evaluated graph $\mathcal{G} _{q _j}$ is the union of the subgraphs from all previous tasks ($\mathcal{G} _{q _j}=\bigcup _{z=1}^{j}\mathcal{G} _{s _{z}}$), and this cumulative test graph includes the inter-task edges connecting these subgraphs into a whole structure. This setup ensures the model cannot utilize the distribution of isolated task graphs to predict task IDs, thereby validating its true continual learning performance6.
>
> We apologize for the confusion we have brought to you, and will commit to clarifying this distinction in greater depth in the revision.
>
> [1] Zhang, Xikun, Dongjin Song, and Dacheng Tao. "Cglb: Benchmark tasks for continual graph learning." Advances in Neural Information Processing Systems 35 (2022): 13006-13021.
>
> > W4. The removal of the limited-representation classes appears to be a strong design choice that requires a more thorough discussion, particularly regarding its advantages and potential drawbacks.
>
> Thank you for your insightful comments. First, we would like to clarify that in our experiments, the class emoval was minimal: we only removed classes with extremely few samples (e.g., classes with less than 10 samples) in the Products and Arxiv-23 datasets. Such extreme sparsity is inherently unsuitable for stable metric calculation and would negatively skew results analysis.
>
> Second, we would like to argue that the motivation behind this setting is to establish a clear benchmark for continual learning and the catastrophic forgetting issue. Label imbalance is a very general problem exists in all classification tasks, consequently, in continual learning, if we do not avoid the label imbalance issue, the label imbalance problem will intertwine with GCL, making it difficult to disentangle the root causes of performance degradation. By removing classes with insufficient samples and utilizing a unified sample size, we isolate and focus strictly on the core continual learning challenge.
>
> We acknowledge that analyzing label imbalance is a crucial real-world problem. In future work, we will commit to specifically analyze the impact of class imbalance on continual learning performance.
>
> > W5 & Q3 In Section 3.2, the authors claim that the considered datasets cover a wide range of possible sizes. However, in practice, the investigation is limited to relatively small datasets. Large-dimensional datasets, such as OGN-Products and Reddit, are not considered. Moreover, most of the selected datasets are citation networks. Considering that the goal of the paper is to establish a benchmark for testing models in GCL, the dataset selection appears rather limited.
>
> Thank you for your comments. We acknowledge not utilizing ultra-large datasets like Reddit or the full OGBN-Products  for our benchmark. This decision was strategically necessary to ensure the computational feasibility of performing over 15 baseline comparisons in a consistent manner, particularly as some GLM-based methods (e.g., GraphGPT and LLaGA) are highly memory- and time-consuming. However, to ensure sufficient scale, we would like to clarify that we have included challenging large-dimensional graph subsets, such as Products (approx. 54k nodes) and Arxiv (approx. 169k nodes), which represent a scale increase over classic benchmarks. Regarding domain diversity, except Citation Networks (Cora, Citeseer, Arxiv, Arxiv-23) , LLM4GCL also covers distinct domains, including the Web Link Network (WikiCS) and E-Commerce/Recommendation Systems (Products, Photo).
>
> In the future, we commit to expanding LLM4GCL to larger datasets with a greater variety of domains to enhance the robustness of the benchmark.

---

> ### Author Response · Authors · 2025-11-25
> **Response [Part3]**
>
> > W6 & Q3 The difference between heterophilous and homophilous datasets is not addressed in this work.
>
> Thank you for your insightful suggestions. We acknowledge that our current work does not explicitly address the impact of heterophily on GCL performance, as our primary focus has been to systematically validate the effectiveness of LLMs on more common and general graph datasets. We will commit to including a dedicated investigation into heterophilous datasets in future iterations of the LLM4GCL benchmark and SimGCL analysis.
>
> > W7 & Q1 In the paper, both the definitions of the literature methods and that of the proposed SimCML are not presented clearly, which makes it difficult to follow the flow of the discussion and to evaluate the novelty of the proposed approach. For instance, the discussion of the LLM baseline is very limited, the authors only state that it is nascent and that no methods have yet been specifically designed for CGL tasks. What remains unclear is how the LLM baselines are actually applied in the context of CGL, and consequently how the reported results are obtained.
>
> Thank you for your careful reviewing. Below is the detailed implementation description of the LLM-based models:
>
> * Encoder-Only Models (BERT, RoBERTa): These models process node text attributes, using the final layer's [CLS] token embedding as the classification feature, which is then passed through a two-layer MLP classifier, mirroring traditional GNN node classification.
> * Decoder-Only Model (LLaMA): It receives a prompt text containing the target node's text attributes and the names of all currently learned classes. Prediction is performed by directly generating the category name, which is then robustly matched (via normalization and case conversion) against the ground-truth label set. The detailed design of this prompt is provided in Appendix D.
> * SimpleCIL: SimpleCIL utilizes the RoBERTa backbone but replaces the MLP classification head with a prototype network for prediction.
> * All LLM baselines integrate the LoRA technique for efficient parameter fine-tuning, aiming to prevent drastic parameter shifts during continuous learning.
>
> We apologize for the confusion we have brought to you, and we will include a detailed model deployment and operation manual in the next version to fully clarify the setup of all baseline models.
>
> > W8 & Q4 Regarding the models considered, it is surprising that the authors do not include several of the most promising replay-based approaches, such as CaT, PDGNN, SSM, among others.
>
> Thank you for your insights. Here we would like to clarify that our primary methodological focus in the LLM4GCL benchmark is **strictly on the rehearsal-free constraint**. This constraint is imposed due to critical real-world limitations, as access to previous task data is often prohibited by privacy concerns or storage limitations. Rehearsal-based approaches benefit from revisiting old data, a resource unavailable to methods like SimGCL, directly introduceing an inherent unfairness comparsion between rehearsal-free and rehearsal-based methods. Therefore, we deliberately excluded all replay-based baselines from our systematic evaluation.

---

> ### Author Response · Authors · 2025-11-25
> **Response [Part4]**
>
> > W9 & Q2 Another major issue is the lack of explanation regarding the experimental setting and the validation policy. While the appendix lists the hyperparameters considered, many of them are fixed, and it is not clear how these values were selected. Considering the aim of the paper, this represents a significant limitation. Moreover, the results reported in the tables do not include any measures of variance or standard deviation, raising concerns about whether the experiments were run only once.
>
> Thank you for your careful reviewing. First we would like to clarify that hyperparameter settings were selected primarily through grid-search, focusing on common parameters that impact model performance across datasets, such as learning rates, hidden dimensions, and batch sizes. For other parameters, we followed the best settings recommended in the original papers or other well-established benchmarks[1-2].
>
> Moreover, as for the running times, we confirm that all reported performance metrics are the average results over five independent runs with different random seeds to ensure statistical robustness. We did not include the standard deviation in our paper due to the limnited table space, as our benchmark reports results of various baselines and datasets. We fully acknowledge the importance of model stability consideration and apologize for the ambiguity in our current manuscript. We will release the detailed measures of variance on our public GitHub repository to facilitate full reproducibility in the near further. We will also add more detailed description of our experimental settings in both the main paper and the Appendix.
>
> [1] Li, Yuhan, et al. "Glbench: A comprehensive benchmark for graph with large language models." Advances in Neural Information Processing Systems 37 (2024): 42349-42368.
>
> [2] Wu, Xixi, et al. "When Do LLMs Help With Node Classification? A Comprehensive Analysis." arXiv preprint arXiv:2502.00829 (2025).

---

> ### Author Response · Authors · 2025-11-27
> **We are looking forward to your feedback**
>
> Dear Reviewer Sw4z,
>
> Thank you again for your efforts in reviewing our manuscript and for providing constructive and insightful comments. Given that the rebuttal period remains only less than one week, we would be very grateful if you could briefly confirm whether our clarifications have resolved your main points. Moreover, if you have any remaining ambiguities or further questions persist, we are pleased to provide detailed elaboration.
>
> Thank you once more for your valuable engagement with our work.
>
> Sincerely,
>
> Authors of Submission 1449

---

### Official Review · Reviewer_ZnXD · 2025-11-01

**Soundness:** 2
**Presentation:** 3
**Contribution:** 3
**Rating:** 6
**Confidence:** 3

**Summary:**

This paper presents a systematic and timely investigation into the potential of large language models (LLMs) to mitigate catastrophic forgetting in graph continual learning (GCL). They introduce a more rigorous global testing setup and develop LLM4GCL, a comprehensive benchmark evaluating nine LLM-based and graph-enhanced LLM (GLM) methods across seven text-attributed graph datasets under both node-level class-incremental (NCIL) and few-shot NCIL (FSNCIL) settings. They propose SimGCL, a simple yet effective GLM-based approach that combines graph-structured prompting, LoRA-based instruction tuning, and training-free prototype classification. This method enables LLMs to capture graph structural information while maintaining strong generalization and avoiding parameter updates that induce forgetting. Extensive experiments demonstrate that SimGCL consistently outperforms existing GNN-, LLM-, and GLM-based baselines—achieving up to 20% higher accuracy under rehearsal-free conditions.

**Strengths:**

1. Critical Re-evaluation of GCL Benchmarks:
The paper makes a valuable methodological contribution by identifying and empirically demonstrating a task ID leakage issue in existing Graph Continual Learning (GCL) benchmarks. This flaw—previously overlooked in the community—renders many reported results unreliable. By introducing a corrected global testing setup, the authors establish a fair and realistic evaluation framework for future GCL research.

2. Bridging GCL and Foundation Models:
Conceptually, the paper establishes an important connection between graph continual learning and pretrained foundation models. By demonstrating that LLMs can serve as effective continual learners on graph-structured data when appropriately prompted, the study opens a new research direction for integrating graph reasoning with large-scale language models.

3. Comprehensive and Insightful Experimental Analysis：
The paper provides a detailed and balanced experimental study, including comparisons among GNNs, LLMs, and GLMs across diverse datasets and settings. The analysis offers clear empirical insights—such as why GLMs may underperform and how prototype-based designs and model scaling influence continual learning performance—making the work informative beyond its own method.

**Weaknesses:**

1. Limited Theoretical Justification for SimGCL:
While SimGCL shows strong empirical results, the paper provides little theoretical or analytical grounding for why the combination of graph prompts, LoRA fine-tuning, and prototype-based classification alleviates forgetting. The approach appears largely empirical, and the mechanism behind its robustness (e.g., whether prototype stability or prompt alignment is the key factor) is not formally analyzed. Adding theoretical reasoning or ablation-based evidence would strengthen the scientific depth of the contribution.

2. Overstatement of SimGCL’s generality and simplicity:
The paper claims that SimGCL “greatly alleviates catastrophic forgetting” and “surpasses all existing baselines” under a rehearsal-free constraint. While the reported results do show strong gains (up to 20%), these improvements are dataset-dependent — performance drops significantly on sparse or long-session datasets (e.g., Arxiv-23, FSNCIL). Hence, calling it universally effective may be an overstatement; the method is empirically strong but not universally superior.

3. Assertion of “efficiency” and “low cost”:
The authors emphasize SimGCL’s efficiency due to LoRA-based tuning and prototype inference. However, they provide no runtime, memory, or scaling benchmarks to substantiate this claim. Given that LLM fine-tuning is resource-intensive, this efficiency claim seems qualitative and not empirically validated, representing a mild overstatement.

**Questions:**

Q1. Clarification on the mechanism of forgetting alleviation
You claim that SimGCL “greatly alleviates catastrophic forgetting” through training-free prototype classification. Could you provide concrete evidence (e.g., forgetting curves, representation drift analysis, or embedding similarity metrics) showing that forgetting is truly reduced, rather than simply avoided by freezing parameters?

Q2. Computational efficiency validation
The paper repeatedly highlights SimGCL’s “efficiency,” but no runtime or GPU memory statistics are presented. Could you quantify the actual training and inference cost (in FLOPs, GPU hours, or wall-clock time) compared to other LLM-based and GNN-based baselines?

---

> ### Author Response · Authors · 2025-11-25
> **Response [Part 1]**
>
> > W1. Limited Theoretical Justification for SimGCL: While SimGCL shows strong empirical results, the paper provides little theoretical or analytical grounding for why the combination of graph prompts, LoRA fine-tuning, and prototype-based classification alleviates forgetting. The approach appears largely empirical, and the mechanism behind its robustness (e.g., whether prototype stability or prompt alignment is the key factor) is not formally analyzed. Adding theoretical reasoning or ablation-based evidence would strengthen the scientific depth of the contribution.
>
> |Model|Cora $\overline{\mathcal{A}}$|Cora $\mathcal{A}_N$|Citeseer $\overline{\mathcal{A}}$|Citeseer $\mathcal{A}_N$|WikiCS $\overline{\mathcal{A}}$|WikiCS $\mathcal{A}_N$|
> |:-|:-:|:-:|:-:|:-:|:-:|:-:|
> |Fine-tune|48.7|33.2|50.1|37.0|38.6|15.4|
> |LoRA|65.6|53.8|55.7|31.7|55.5|30.9|
> |LoRA+Proto.|78.5|65.4|70.6|57.6|60.3|54.4|
> |LoRA+Prompt|68.5|55.1|60.3|39.6|63.9|41.7|
> |LoRA+Prompt+Proto.|84.6|80.0|77.1|66.3|73.5|61.9|
>
> Thank you for your suggestions. Following your advice, we conduct an ablation study comparing the contribution of LoRA, graph prompt and prototype, the results are in the above table. It can be observed that, firstly, the performance of the model without LoRA tuning is significantly worse, as the full-parameter tuning will change more LLMs' paprameters during various sessions, leading to more severe catastophic forgetting and model performance degradation. Secondly, the contribution of the prototype mechanism to model performance is substantial. Without the prototype mechanism, the performance gain from graph prompt is limited, because the model still undergoes continuous parameter updates across sessions. However, the improvement becomes significant when the prompt is introduced alongside the prototype. We hypothesize this synergy exists because, although the graph prompt can provide richer information, it intensify the model's tendency toward overfitting. In contrast, by freezing the parameters, prototype network ensures model's stability, which allows the graph prompt to supply the auxiliary information needed for the model to make more accurate judgments in incremental sessions without parameter updation.
>
> We acknowledge that theoretical analysis will help with deeper understanding of the models. However, here we would like to clarify that the primary goal of this paper is to establish the first "LLM for GCL" benchmark and propose an empirically effective method SimGCL for this scenario. In the future, we will focus on making formal theoretical analysis of graph prompts and prototype networks, to make this work more rigious and solid.
>
> > W2. Overstatement of SimGCL’s generality and simplicity: The paper claims that SimGCL “greatly alleviates catastrophic forgetting” and “surpasses all existing baselines” under a rehearsal-free constraint. While the reported results do show strong gains (up to 20%), these improvements are dataset-dependent — performance drops significantly on sparse or long-session datasets (e.g., Arxiv-23, FSNCIL). Hence, calling it universally effective may be an overstatement; the method is empirically strong but not universally superior.
>
> Thank you for your assessments. Firstly, we acknowledge that the performance description in the Introduction was kind of overstated. We will revise and temper this absolute claim in the next version manuscript.
>
> However, we also want to argue that SimGCL's success is not dataset-specific. It consistently ranks first or second across the majority of NCIL and FSNCIL comparisons. Furthermore, our extended experiments confirm SimGCL's SOTA results in the Domain-Incremental Learning setting (Appendix E.3, Table 11). This broad effectiveness strongly validates our model's adaptability across multiple datasets and distinct CL scenarios.
>
> Moreover, we also want to emphasize that SimGCL is a flexiable framework, which can be enhanced for these specific scenarios through targeted modifications, like utilizing prototype calibration techniques to balance class representations while stabilizing prototype evolution in long-session datasets, or implementing conditioned, high-order graph structure information for effective processing of sparse graphs. These are primary directions for our future work.

---

> ### Author Response · Authors · 2025-11-25
> **Response [Part 2]**
>
> > Q1. Clarification on the mechanism of forgetting alleviation. You claim that SimGCL “greatly alleviates catastrophic forgetting” through training-free prototype classification. Could you provide concrete evidence (e.g., forgetting curves, representation drift analysis, or embedding similarity metrics) showing that forgetting is truly reduced, rather than simply avoided by freezing parameters?
>
> Thank you for your insightful comments. We would like to clarify that we have included two experiments reporting the embedding visualization and training curves in **Appendix E Figure 4 and Figure 5**, respectively.
>
> In Figure 4, we performed a visualization experiment comparing the embeddings and prototypes generated by three prototype-based methods: Cosine, SimpleCIL, and SimGCL. It can be observed that SimGCL's learned embeddings exhibit the largest inter-class distances and smallest intra-class variance, with samples tightly clustered around their respective prototypes. These findings underscore the superior embedding capability of
> SimGCL, reflecting its enhanced ability to comprehend both graph structures and textual information.
>
> In Figure 5, we report the performance evolution curves of all models across continuous sessions. The decline slope (i.e., the forgetting rate) of SimGCL is significantly less pronounced than those of GNN-based and other LLM-based, GLM-based baselines, which provides direct empirical evidence that forgetting is substantially and effectively reduced.
>
>
> > W3 & Q2. Computational efficiency validation The paper repeatedly highlights SimGCL’s “efficiency,” but no runtime or GPU memory statistics are presented. Could you quantify the actual training and inference cost (in FLOPs, GPU hours, or wall-clock time) compared to other LLM-based and GNN-based baselines?
>
> Thank you for your careful reviewing. We would like to point out that we have already conducted a detailed time efficiency comparison with GNN-, LLM-, and GLM-based baselines, with results presented in **Appendix E.4, Table 12**. The results demonstrate that SimGCL is more efficient than other LLaMA-based methods (LLaMA, GraphPrompter, GraphGPT, LLaGA) in terms of model size, training time, and inference time. While
> its computational cost exceeds traditional GNN methods, this is offset by its superior performance. Moreover, SimpleCIL also achieves competitive efficiency, presenting running-up performance with computational costs similar to the GNN-based method TPP. These results suggest that well-designed LLM approaches can balance performance and computational cost effectively.

---

> ### Author Response · Authors · 2025-11-27
> **We are looking forward to your feedback**
>
> Dear Reviewer ZnXD,
>
> Thank you again for your efforts in reviewing our manuscript and for providing constructive and insightful comments. Given that the rebuttal period remains only less than one week, we would be very grateful if you could briefly confirm whether our clarifications have resolved your main points. Moreover, if you have any remaining ambiguities or further questions persist, we are pleased to provide detailed elaboration.
>
> Thank you once more for your valuable engagement with our work.
>
> Sincerely,
>
> Authors of Submission 1449

---

### Meta-Review · Area_Chair_DCqf · 2025-12-08

**Summary:**

The paper investigates whether pretrained LLMs and graph-LLMs can mitigate catastrophic forgetting in graph continual learning (GCL). It identifies “task ID leakage” in common GCL evaluations and proposes a stricter global testing protocol, introduces the LLM4GCL benchmark spanning multiple LLM/GLM/GNN baselines across seven text-attributed graph datasets, and presents SimGCL, which combines graph-structured prompting, a one-time LoRA-based instruction-tuning stage, and training-free prototype classification in later sessions.


Before the rebuttal, reviewers’ concerns focused on: (i) clarity and completeness of the method, baselines, and experimental protocol—including how LLM baselines are instantiated, how hyperparameters and validation are handled, and the absence of variance reporting; (ii) fairness and motivation—whether computationally expensive LLMs are appropriate for a setting motivated by efficiency, whether comparisons across heterogeneous backbones are fair, and whether TAG datasets overlap with LLM pretraining corpora; and (iii) scope and generality—limited dataset diversity (heavily citation-network–based), missing heterophily analysis, unclear extensibility beyond TAGs, and requests for stronger ablations and theoretical grounding.

The rebuttal partially addresses these issues, but several core concerns remain. First, technical clarity still needs substantial improvement, particularly in articulating baseline design choices, experimental assumptions, and methodological rationale. Second, the motivation and the method’s design are not well aligned: the introduction positions the paper’s contribution around flaws in existing GCL evaluation protocols, yet it remains unclear how these setup issues concretely motivate or necessitate the architectural choices in SimGCL. Third, although the paper claims to address catastrophic forgetting in graph continual learning, the method’s essence aligns closely with template-based or in-context style learning, especially given its reliance on a frozen backbone and prototype-based classification. The connection between this mechanism and the core continual-learning objective—mitigating representational drift and preserving knowledge across sessions—needs to be more rigorously established. The absence of such grounding leaves a conceptual gap between the stated problem and the proposed solution.

**Reviewer Concerns:**

Across reviews, the rebuttal successfully clarified several factual and experimental details, but a number of core conceptual and methodological concerns remain unresolved.

**Addressed concerns.**
The rebuttal provided clearer descriptions of: (i) the baseline implementations and training protocols; (ii) the global-testing setup and how inter-task edges and class sets are handled across sessions; (iii) ablation evidence on the roles of LoRA, prompts, and prototypes; and (iv) dataset construction choices, sample-size filtering, and variance reporting (confirming five-seed averages). It also supplied pointers to additional visualizations (embedding clustering, forgetting curves) and time-efficiency analyses in the appendix. These clarifications resolve many presentation-level issues raised by reviewers.

**Outstanding concerns.**
However, several fundamental concerns remain.

* Technical clarity and coherence of the method. Despite added explanations, the paper still does not clearly motivate and justify its design choices. The link between the identified evaluation issues (e.g., task-ID leakage) and the specific components of SimGCL (graph-prompt verbalization, one-time LoRA tuning, prototype classifier) is not made conceptually tight.

* Motivation–method mismatch. Reviewers noted that the introduction frames the main contribution as correcting unfair or flawed GCL evaluation protocols, but the proposed method largely operates at the architectural/representation level. The rebuttal does not fully establish how the diagnosis of protocol flaws directly leads to the design of SimGCL.

**Reviewer Scores:**

The rebuttal successfully resolves several presentation and implementation concerns, but the more fundamental conceptual and methodological issues—particularly the alignment between the stated motivation and the proposed method, as well as the theoretical grounding of the approach—remain largely unaddressed.

Given the improved clarity and added explanations, I expect Reviewer Sw4z would likely raise their score from 2 to 4. For the remaining reviewers, the rebuttal does not sufficiently change the core concerns they identified, so their original scores are likely to remain unchanged.

---

### Decision · Program_Chairs · 2026-01-26

Reject